# *"A Tale of Two Movements"*: Identifying and Comparing Perspectives in #BlackLivesMatter and #BlueLivesMatter Movements-related Tweets using Weakly Supervised Graph-based Structured Prediction

**Shamik Roy**[*]
Purdue University
roy98@purdue.edu

**Dan Goldwasser**
Purdue University
dgoldwas@purdue.edu

## Abstract

Social media has become a major driver of social change, by facilitating the formation of online social movements. Automatically understanding the perspectives driving the movement and the voices opposing it, is a challenging task as annotated data is difficult to obtain. We propose a weakly supervised graph-based approach that explicitly models perspectives in #BackLivesMatter-related tweets. Our proposed approach utilizes a social-linguistic representation of the data. We convert the text to a graph by breaking it into structured elements and connect it with the social network of authors, then structured prediction is done over the elements for identifying perspectives. Our approach uses a small seed set of labeled examples. We experiment with large language models for generating artificial training examples, compare them to manual annotation, and find that it achieves comparable performance. We perform quantitative and qualitative analyses using a human-annotated test set. Our model outperforms multitask baselines by a large margin, successfully characterizing the perspectives supporting and opposing #BLM.

## 1 Introduction

Social media platforms have given a powerful voice to groups and populations demanding change and have helped spark social justice movements. The platforms provide the means for activists to share their perspectives and form an agenda, often resulting in real-world actions. Such movements can also lead to the formation of reactionary movements, aiming to counter the call for change. In this paper we suggest a computational framework for analyzing such instances of public discourse, specifically looking at the interaction between *#BlackLivesMatter* and its reaction - *#BlueLivesMatter*.

The first movement was formed in 2013 in response to the acquittal of George Zimmerman

---

[*]The work was done when the first author was at Purdue University.

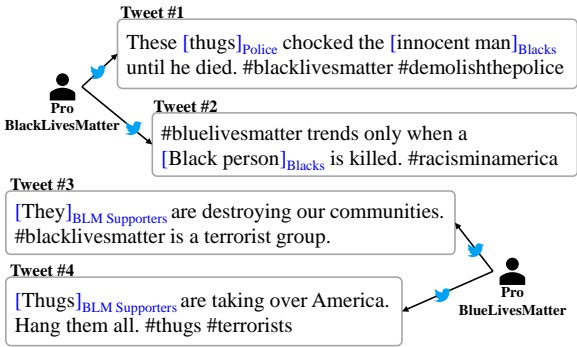

Figure 1: Author stances help disambiguate entity references and understanding perspectives.

of killing of Trayvon Martin, an unarmed Black teenager. While support for the movement fluctuated (Parker et al., 2020) and varied across different demographic groups (Horowitz and Livingston, 2016), since May 2020, following the murder of George Floyd, support has increased across all U.S. states and demographic groups (Kishi and Jones, 2020; Parker et al., 2020), changing public discourse (Dunivin et al., 2022) and resulting in widespread protest activities (Putnam et al., 2020). At the same time, these activities have attracted counter movements (Gallagher et al., 2018; Blevins et al., 2019). While *#BlackLivesMatter* protested against police violence towards Black individuals, the counter movement, referred to using its most prominent hashtag, *#BlueLivesMatter*, emphasized the positive role and need for law enforcement, often painting protesters as violating the law.

To account for the differences in *perspectives* between the two movements in a nuanced way, we suggest a structured representation for analyzing online content, corresponding to multiple tasks, such as capturing differences in stances, disambiguating the entities each side focuses on, as well as their role and sentiment towards them. We discuss our analysis in Section 2.2. From a technical perspective, our main challenge is to model the dependency among these tasks. For example, con-

| | |
|---|---|
| # Authors | 31,704 |
| # Retweet relations | 3,206 |
| # Keywords in author profiles | 9,500 |
| # Tweets | 402,647 |
| # Entity mentioned in tweets | 393,441 |
| # Hashtags used in tweets | 1,068,525 |
| Time span | 05/26/20 to 06/26/20 |

Table 1: Unlabeled #BLM corpus statistics.

sider the use of the entity "*thugs*" in Figure 1 by two different authors. Understanding the perspectives underlying the texts' *stance* towards *#BlackLives-Matter*, requires *disambiguating* it as either refering to police officers, or to BlackLivesMatter supporters. We model such interaction among tasks using a graph-based representation, in which authors, their posts and its analysis are represented as nodes. We use a graph neural network (Schlichtkrull et al., 2018) to model the interaction among these elements, by creating graph-contextualized node representations, which alleviate the difficulty of text analysis. We discuss our graph representation in Section 3.1 and its embedding in Section 3.2.

Social movements are dynamic, as a result, using human-annotated data for training is costly. Hence, we explore learning from weak supervision, initialized with artificially generated data using Large Language Models (LLMs) (Brown et al., 2020) and amplified using self-training, exploiting the graph structure to impose output consistency. We discuss self-learning in Section 3.5.

To evaluate our model and the quality of artificial data, we human annotate $\sim 3k$ tweets by both sides for all analysis tasks. We compare our graph-based approach with multitask and discrete baselines in two settings - direct supervision (train with human-annotated data) and weak supervision (train with artificial data). Our results on human-annotated test set consistently demonstrate the importance of social representations for this task and the potential of artificially crafted data. We also present an aggregate analysis of the main perspectives on $400k$ tweets related to the George Floyd protests.

## 2 BLM Data Collection and Analysis

First, we describe the data collection process for the #BLM movement. Then, we define the structured perspective representation used for analyzing it.

### 2.1 Dataset

For studying perspectives on the BLM movement we use the dataset collected by Giorgi et al., 2022. This dataset contains tweets in various languages

| Abstract Entities | Common Perspectives in | |
|---|---|---|
| | Pro BlackLM | Pro BlueLM |
| **Black Americans** | Positive Target | Negative Actor |
| **Police** | Negative Actor | Positive Actor, Positive Target |
| **Community** | N/A | Positive Target |
| **Racism** | Negative Actor | N/A |
| **Democrats** | N/A | Negative Actor |
| **Republicans** | N/A | Positive Actor |
| **Government** | Negative Actor | N/A |
| **White Americans** | Negative Actor | N/A |
| **BLM Movement** | Positive Actor, Positive Target | Negative Actor |
| **Petition** | Positive Target | N/A |
| **Antifa** | N/A | Negative Actor |

Table 2: Common abstract entities and perspectives in BlackLM and BlueLM campaigns. Only 29% entities in the corpus are covered using an exact lexicon match approach for entity disambiguation.

on the BLM protest and the counter-protests. This dataset was collected using keyword matching and spans from 2013 to 2021. However, as shown in the paper, BLM-related tweets spiked mainly after the murder of George Floyd in 2020. Hence, in this paper, we study the tweets that were posted in a month time span following George Floyd's murder. We consider original tweets written in English and discard any author from the dataset who tweeted less than 5 times in the timeframe. The dataset statistics can be found in Table 1.

### 2.2 Defining Perspectives

**Capturing Perspectives.** Previous study (Giorgi et al., 2022) has focused on topic indicator keywords for understanding differences in perspectives among BLM and counter-movements. However topic indicator lexicons might not capture the intended meaning. For example, the intent of hashtag use is often ambiguous, as shown in Figure 1, both authors use '#blacklivesmatter' to express opposite perspectives. Previous studies (Rashkin et al., 2016; Roy et al., 2021; Pacheco et al., 2022) have shown that sentiment toward entities can disambiguate stances in polarized topics, we follow this approach and adapt the Morality Frames proposed by Roy et al., 2021 that are linguistic frames that capture entity-centric moral foundations. Adapting this work, we use four dimensions of perspectives towards a specific entity - actor ("do-er", having agency) and target ("do-ee", impacted by the actions of the actor), and sentiment, positive or negative, based on the author's view (e.g., negative

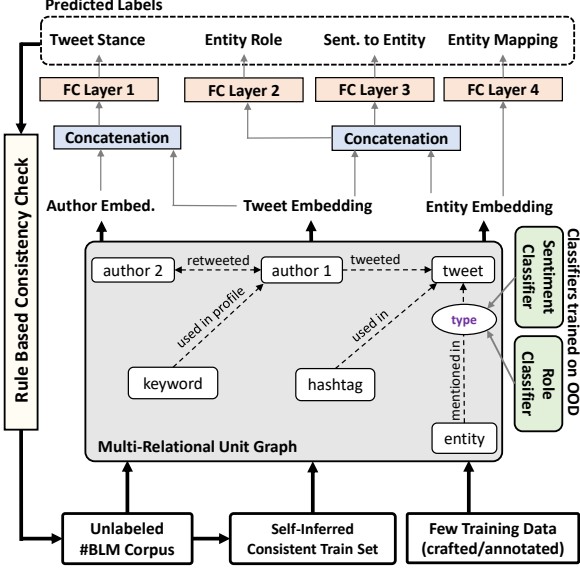

Figure 2: Proposed self-supervised model.

actor, will have a negative impact on the target). We characterize the moral reasoning behind that sentiment using Moral Foundation Theory (Haidt and Joseph, 2004; Haidt and Graham, 2007), however, given the difficulty of learning abstract moral dimensions without direct supervision, we rely on external tool and report the results at an aggregate.

**Entity Disambiguation.** The focus on specific entities is central to the movements' messaging. However, identifying it directly from text can be challenging. For example, in Figure 1, 'thugs' map to both 'police' or 'BLM Supporters' depending on the speaker. To alleviate this difficulty, we identify the key *abstract entities* discussed by both sides using human-in-the-loop data analysis at an aggregate level (listed in Table 2). The description of abstract entities and the procedure for identifying them can be found in Appendix A. We define an entity disambiguation task, mapping the raw entities that appear in tweets to the intended abstract entity.

**Stance.** We study the perspectives from two opposing standpoints - Pro-BlackLivesMatter (addressed as pro-BlackLM) and Pro-BlueLivesMatter (addressed as pro-BlueLM).

# 3   Weakly Supervised Identification of Perspectives

In this section, we propose a self-supervised modeling approach for a holistic identification of perspectives and stances on #BlackLivesMatter.

## 3.1   Structured Representation of Text

To jointly learn perspectives and identify stances on #BLM, we define perspectives as how entities are portrayed by an author as described in Section 2. To identify perspectives towards an entity in a tweet we need to identify - (1) the abstract entity the targeted entity maps to, and (2) the role assigned and sentiment expressed towards the entity. Our insight is that the entity mapping, entity role/sentiment, and author stance all are interdependent decisions. For example, if an entity maps to the abstract entity "police" and the author's stance is "pro-BlackLM", then the sentiment towards the entity will most likely be "negative" and the entity role will be "actor". Similarly, if an author is identified to be repeatedly mentioning "police" as a negative actor, then an unknown entity mentioned by the same author, that is identified to have a role of a "negative actor" will most likely address "police". To explicitly model these dependencies between authors' stances and perspectives, we first convert the large unlabeled tweet corpus (in Table 1) to a graph consisting of textual elements (text, entity mentions, hashtags). Then we connect the textual graph to the social context (author retweet graph). A unit representation of our multi-relational graph is shown in Figure 2 and various nodes and relations among them in this graph are described below.

**(1) Author-tweets-Tweet:** An author node is connected to the tweet nodes they tweeted.
**(2) Author-retweets-Author:** Author nodes can be connected to each other with a retweet relationship.
**(3) Author-uses-Keyword:** An author node can be connected to a keyword node that is mentioned in their profile description. Keywords are meaningful hashtags and hashtag-like phrases. For example, "Black Lives Matter" is a phrase, and "#blacklivesmatter" is a similar hashtag.
**(4) Hashtag-used-in-Tweet:** A hashtag node is connected to the tweet node it is used in.
**(5) Entity-mentioned-in-Tweet:** An entity node is connected to the tweet node it is mentioned in.

## 3.2   Learning Representation using GCN

After converting text to graph and connecting it with the social network of authors, we learn the representation of the graph elements using a multi-relational Graph Convolutional Network (R-GCN) (Schlichtkrull et al., 2018), an adaptation of traditional GCN (Kipf and Welling), where activations

from neighboring nodes are gathered and transformed for each relation type separately. The representation from all relation types is gathered in a normalized sum and passed through an activation function (ReLU) to get the final representation of the node. A 2-layer R-GCN is sufficient in our case to capture all dependencies, resulting in a rich-composite representation for each node.

### 3.3 Predictions on Learned Representation

We define a set of prediction tasks on the learned representation of the nodes using R-GCN. These prediction tasks help infer various labels for the graph elements and at the same time maintain consistency among various types of decisions. In the following objectives, $E$ represents learnable representation of a node using R-GCN, $H$ represents trainable fully connected layers, $\sigma$ represents Softmax activation, and $\oplus$ means concatenation.

**Tweet stance prediction:** Intuitively, the stance in a tweet depends on its author. Hence, for a tweet, $t$, and its author, $a$, we define the classification task of tweet stance (pro-BlackLM/pro-BlueLM) as follows which ensures consistency among author stances and their generated tweet stances.

$$\hat{y}_{tweet-stance} = \sigma(H_{tweet-stance}(E_a \oplus E_t))$$

**Entity sentiment, role, and abstract entity label prediction:** The sentiment and role of a given entity depend on the tweet text it is mentioned in. Given an entity, $e$, and the corresponding tweet, $t$, we define the classification task of the entity sentiment (pos./neg.) and role (actor/target) as follows.

$$\hat{y}_{sent} = \sigma(H_{sent}(E_e \oplus E_t))$$
$$\hat{y}_{role} = \sigma(H_{role}(E_e \oplus E_t))$$

Intuitively, the abstract entity label does not directly depend on the tweet text rather it depends mostly on its textual properties and indirectly on the sentiment/role assigned to it. We define the abstract entity label (out of 11 entities in Table 2) prediction by conditioning only on the entity representation.

$$\hat{y}_{map} = \sigma(H_{map}(E_e))$$

Additionally, to maintain consistency between stance and perspectives towards entities (e.g. pro-BlackLM and "police neg. actor" are consistent), we define a prediction task of stance on the learned representation of entities and tweets as follows.

$$\hat{y}_{entity-stance} = \sigma(H_{entity-stance}(E_e \oplus E_t))$$

### 3.4 Entity Role and Sentiment Priors

The "mentioned-in" relationship between an entity and the tweet it is mentioned in, may have multiple types e.g., pos-actor, neg-actor, and pos-target, based on the sentiment expressed and the role assigned to this entity. We initialize the priors of this relationship type in our graph using two off-the-shelf classifiers, $C_{sent}$ and $C_{role}$, for sentiment and role classification, respectively, trained on out-of-domain data (OOD). They are defined as follows.

$$C_{sent/role}(E_e^0) = \sigma(H'_{sent/role}(E_e^0))$$

We align predictions from $C_{sent}$, $C_{role}$ (on the large unlabeled corpus) to the "mentioned-in" edges using the following loss functions.

$$L_{sa} = m\left(\sigma(H_{sent}(E_e \oplus E_t)), C_{sent}(E_e^0)\right)$$
$$L_{ra} = m\left(\sigma(H_{role}(E_e \oplus E_t)), C_{role}(E_e^0)\right)$$

Here, $E_e^0$ is the non-trainable input representation of an entity e and $m$ represents L1 loss. These learning objectives set up the priors for entity sentiment and role in our framework. In course of training, these priors get updated as the parameters of the classifiers $C_{sent}$, $C_{role}$ are also updated.

We define the final loss function $L$ as the sum of the alignment losses, $L_{sa}$ and $L_{ra}$ and all prediction losses described above, generally denoted as $P(\hat{y}, y)$ where $\hat{y}$ is the predicted label, $y$ is the gold label, and $P$ is CE loss. We optimize $L$ by updating all trainable parameters including R-GCN.

### 3.5 Self Learning Loop

We propose a self-learning procedure where our model checks consistency among all predictions after every $k$ learning steps which we name the "Inference step". The model adds elements (e.g., tweet, entity) with consistent labels found in this step, to its training set for the next iterations. In this manner, our model is able to start with only a few annotated examples as a training set and keep annotating data and increasing training examples in a continuous learning loop. We permit $k$ epochs after each inference step, $I_j$, so that the model parameters are learned based on the new training examples inferred at the previous inference step, $I_{j-1}$. The model identifies consistent labels using the following consistency checks sequentially.

**(1) Label confidence:** A label is reliable if predicted with a minimum confidence of $c \in [0, 1]$. Only reliable labels are used in the next checks.

```
<prompt-start>
Write 30 different tweets that must -
1. Explicitly mention one or more entities from the following list [Black man, Black people, Black woman, Black
community] and portray them as positive targets.
2. Be in the support of the "Black Lives Matter" movement.
3. Use one or more hashtags from the following list: [#blacklivesmatter, #blm, #policebrutality, #racism].

Answer:
<prompt-end>

<generation-start>
1. I stand with the black community and the "Black Lives Matter" movement. #blacklivesmatter #blm
2. We must continue to fight for justice and equality for all black people. #alllivesmatter #policebrutalitypandemic
3. No one should have to live in fear because of the color of their skin. #blacklivesmatter #racism
4. All black lives matter - from the oldest to the youngest. We must protect them all. #blm #policebrutality
5. Black women are the backbone of our community and we must support them always. #blacklivesmatter #blm
6. We must work together to end the systematic racism and oppression of black people. #blacklivesmatter #racism
7. We will no longer tolerate the unjust murder of black people by those in positions of power. #blm
8. It's time for a change - we need to see more black people in leadership positions. #blacklivesmatter
9. The fight for racial equality is not over until all black lives matter. #alllivesmatter #policebrutality
10. We must come together to stand up against racism and prejudice in all forms. #blacklivesmatter
......
......
```

Figure 3: Prompt example for GPT-3 to generate tweets having "pro-BlackLM" stance and portraying "Black Americans" as "positive target". The black-colored text is the input text in the prompt and the blue-colored texts are generated tweets by GPT-3. The generic prompt structure can be observed in Figure 7 in Appendix B.

**(2) Tweet consistency:** A tweet is consistent if the predicted stance of the tweet is consistent with the perspectives identified towards all mentioned entities in the tweet (determined from Table 2). For example, a tweet stance pro-BlackLM is consistent with a mentioned entity in the tweet identified as "police-neg-actor" and inconsistent with an entity predicted as "police-pos-target".

**(3) Author consistency:** An author is consistent if all of their consistent tweets (identified in (2)) are labeled to have the same stance (pro-BlackLM or pro-BlueLM) and they have at least $t$ such tweets.

Tweets from the consistent authors and the corresponding consistent entities are added to the training set for the next learning steps. We also keep training the off-the-shelf sentiment and role label classifiers ($C_{sent}$, $C_{role}$) at each step, using the inferred training set by our model so that they are up-to-date. We keep running our model until the model does not generate more than $y\%$ new training data for consecutive $x$ inference steps.

## 4 Training and Test Data Collection

Detecting perspectives in social media texts using supervised models requires costly human-annotated training data. Hence, being inspired by the recent advances in generative Large Language Models (LLMs) (Min et al., 2021), we artificially craft training examples using LLMs. To evaluate the quality of the crafted training data and to evaluate our model's performance in real test data, we human annotate a subset of real data.

**Artificial Training Data Generation:** For generating tweets containing specific perspectives, we prompt LLMs (Brown et al., 2020) in a way that all of the structured elements for the perspectives are present in the generated tweets. For example, to generate tweets that are "pro-BlackLM" and portray "Black Americans" as "positive target", we prompt LLMs to generate $N$ tweets that meet 3 conditions - (1) explicitly mention one or more entities from ('Black man', 'Black people', 'Black woman', etc.) and portray them as positive targets, (2) are in the support of the "Black Lives Matter" movement, (3) use one or more hashtags from (#blacklivesmatter, #blm, etc.). We follow this prompt structure for each (stance, abstract entity, perspective) tuple in Table 2. We find that the LLM-generated tweets are pretty realistic. An example prompt and some generated examples are shown in Figure 3. The generic prompt structure can be observed in Figure 7 in Appendix B. We convert the generated artificial training examples to graph structures as before. For each (stance, abstract entity, perspective) pair we construct an imaginary author node in the graph whose embeddings are initialized by averaging the corresponding artificial tweets. In this manner, we get the same unit structure for the artificial tweets as the real tweets. We experiment with two LLMs, GPT-3 (Brown et al., 2020) and GPT-J-6B (Wang and Komatsuzaki, 2021). We observed that GPT-J generated mostly repetitive examples, hence, we use the GPT-3 generations. Detailed prompting methods, generated data statistics and examples, LLM

hyperparameters, and pre/post-processing steps can be found in Appendix B.

**Human Annotation of Real Data:** To evaluate the quality of the artificially crafted data and to evaluate our model's performance in real data, we human annotate a subset of the real data. We sample 200 authors from the corpus described in Table 1 and human-annotate these authors and their tweets for stances (pro-BlackLM/pro-BlueLM). We also human-annotate the entities they mention in their tweets for sentiment towards them (pos./neg.), their role (actor/target), and their mapping to the abstract entities (one of the 11 abstract entities in Table 2). Each data point is annotated by two human annotators and we find substantial to almost perfect agreement in all the annotation tasks. We resolve disagreements between the two annotators in all of the annotation steps by discussion. We observe that often the supporters of the movements "hijack" (Gallagher et al., 2018) the opponent's hashtags to troll or criticize them. We annotate these tweets as "Ambiguous Tweets" and are identified by looking at keyword usage (e.g., when a pro-blackLM tweet uses the keyword "bluelivesmatter"). Detailed human annotation process, inter-annotator agreement scores, examples of ambiguous tweets, and per-class data statistics are in Appendix C.

| | | # of Authors | # of Tweets | # of Ambiguous Tweets | # of Entities |
|---|---|---|---|---|---|
| TRAIN | LLM (GPT-3) Generated (Weak Supervision) | - | 582 | - | 517 |
| | Human Annotated (Direct Supervision) | 50 | 721 | 242 | 444 |
| TEST | Human Annotated | 139 | 2259 | 278 | 1647 |

Table 3: Training and test data statistics.

**Train-Test Split:** We randomly sample a small subset of the human-annotated authors and use these authors, their tweets, and mentioned entities as training set and the rest of the data as a test set for our proposed model and all baselines. We perform our experiments in two setups - **(1) Weak supervision:** LLM-generated examples are used for training, **(2) Direct supervision:** human-annotated real data are used for training. In both setups, the models are tested on the human-annotated test set. The statistics for the LLM-generated train set and human-annotated train/test set are shown in Table 3. Note that our self-learning-based model depends only on a few training examples for initial supervision, hence, few training data in both settings are enough to bootstrap our model.

## 5 Experimental Evaluation

### 5.1 Experimental Settings

We first perform task adaptive pretraining (Gururangan et al., 2020) of RoBERTa (Liu et al., 2019) using unused #BLM tweets from the dataset by Giorgi et al., 2022 (addressed as RoBERTa-tapt). We use it for implementing the baselines and initializing the nodes in our model. We observe improvement in tasks using RoBERTa-tapt over basic RoBERTa (in Table 4). We use RoBERTa-based classifiers as external classifiers, $C_{sent}$ and $C_{role}$, and pretrain them on out-of-domain data proposed by Roy et al., 2021. Details of our model initialization, hyperparameters, stopping criteria, pretraining $C_{sent}$ and $C_{role}$, etc. are in Appendix D.3.

**Baselines:** Our task of the joint identification of perspectives in terms of sentiment toward main actors, their roles, their disambiguation, and stance identification, is unique and to the best of our knowledge, no other works studied unified perspective detection in such a setting. Hence, the closest baseline of our model is the multitask modeling approach (Collobert and Weston, 2008). We compare our model with a RoBERTa-based multitask approach where a shared RoBERTa encoder is used and fine-tuned for the identification of entity sentiment, role and mapping, and tweet stances. To compare it with our proposed model that incorporates author social-network interaction, we enhance the multitask RoBERTa by concatenating social-network-enhanced author embeddings with text representations. We also compare with discrete RoBERTa-based text classifiers where all tasks are learned separately by fine-tuning a separate RoBERTa encoder for each task. We also compare with the keyword-matching-based baseline for stance classification (Giorgi et al., 2022). For a fair comparison, we pretrain all sentiment and role classification baselines on the same out-of-domain data by Roy et al., 2021 that we use for pertaining $C_{sent}$ and $C_{role}$. Details of the baselines and their hyperparameters are in Appendix D.4.

### 5.2 Results

We present all individual classification results in Table 4. Our first observation is that the identification of stances in the ambiguous tweets is difficult as, by definition, they "hijack" (Gallagher et al., 2018) the opponent's keywords. As a result, the keyword-based baseline performs best among all baselines in overall tweet stance classification, however, it

| | AUTHOR STANCE | | ALL TWEET STANCE | | AMB. TWEET STANCE | | ENTITY SENTIMENT | | ENTITY ROLE | | ENTITY MAPPING | |
| MODELS | Weak Sup. | Direct Sup. | Weak Sup. | Direct Sup. | Weak Sup. | Direct Sup. | Weak Sup. | Direct Sup. | Weak Sup. | Direct Sup. | Weak Sup. | Direct Sup. |
|---|---|---|---|---|---|---|---|---|---|---|---|---|
| **NAIVE** | | | | | | | | | | | | |
| Random | 50.2 ± 1.9 | | 48.42 ± 0.5 | | 49.19 ± 1.9 | | 50.07 ± 1.6 | | 49.49 ± 1.4 | | 7.34 ± 0.5 | |
| Keyword Based | 88.82 ± 1.0 | | 87.77 ± 0.4 | | 21.57 ± 1.3 | | - | | - | | - | |
| **DISCRETE** | | | | | | | | | | | | |
| RoBERTa | 70.48±10.0 | 78.19 ± 6.6 | 66.78 ± 6.3 | 73.63 ± 3.4 | 27.93 ± 3.1 | 54.18 ± 4.7 | 76.45 ± 1.6 | 80.57 ± 1.3 | 74.76 ± 0.6 | 82.53 ± 1.5 | 43.59 ± 3.6 | 53.21 ± 6.0 |
| RoBERTa-tapt | 77.58±11.6 | 86.23 ± 2.6 | 75.49 ± 9.8 | 82.69 ± 1.9 | 31.61 ± 1.6 | 68.03 ± 6.3 | 84.31 ± 1.5 | 86.17 ± 0.2 | 84.53 ± 1.1 | 86.25 ± 0.5 | 47.62 ± 2.8 | 45.71 ± 4.2 |
| **MULTITASK** | | | | | | | | | | | | |
| RoBERTa | 74.65 ± 5.6 | 82.51 ± 3.8 | 67.08 ± 5.5 | 78.39 ± 2.9 | 29.32 ± 1.8 | 55.08 ± 4.4 | 76.99 ± 0.8 | 79.18 ± 0.7 | 74.42 ± 1.7 | 83.7 ± 1.0 | 52.4 ± 1.6 | 47.76 ± 6.2 |
| RoBERTa-tapt | 79.57 ± 3.9 | 90.26 ± 1.6 | 76.69 ± 4.7 | 86.25 ± 1.2 | 32.25 ± 1.2 | 73.12 ± 2.1 | 84.83 ± 1.1 | 86.29 ± 0.3 | 83.69 ± 0.6 | 87.33 ± 0.6 | 52.62 ± 3.2 | 44.35 ± 3.3 |
| + Author Embed. | 81.81 ± 1.4 | 90.48 ± 2.0 | 76.03 ± 2.2 | 85.08 ± 1.0 | 31.58 ± 0.7 | 71.76 ± 4.8 | 85.17 ± 1.1 | 86.79 ± 0.4 | 83.03 ± 0.8 | 87.17 ± 0.2 | 49.93 ± 2.2 | 46.42 ± 3.5 |
| **OUR MODEL** | | | | | | | | | | | | |
| Text-discrete | 71.78 ± 8.7 | 78.65 ± 4.0 | 69.12 ± 7.8 | 65.9 ± 4.4 | 32.16 ± 2.1 | 62.88 ± 4.5 | 83.31 ± 0.9 | 83.76 ± 0.6 | 84.14 ± 0.9 | 83.28 ± 0.9 | 35.84 ± 1.9 | 15.47 ± 9.5 |
| Text-as-Graph | 79.36 ± 3.3 | 77.78 ± 2.5 | 78.25 ± 3.0 | 78.15 ± 2.0 | 34.96 ± 5.2 | 43.8 ± 2.2 | 84.86 ± 0.2 | 85.29 ± 0.5 | 85.92 ± 0.2 | 85.83 ± 0.4 | 49.83 ± 3.1 | 61.14 ± 2.1 |
| + Author Network | 82.48 ± 2.2 | 93.28 ± 1.1 | 89.72 ± 1.6 | 95.66 ± 0.8 | 35.11 ± 0.1 | 87.23 ± 4.0 | 84.92 ± 0.2 | 84.65 ± 0.5 | 86.0 ± 0.2 | 85.79 ± 0.3 | 49.42 ± 3.6 | **62.74±2.0** |
| + Self-Learning | **90.83±1.2** | **95.39±1.9** | **93.37±0.5** | **95.85±2.8** | **63.43±5.5** | **92.47±4.6** | **87.02±0.5** | **87.63±0.4** | **86.72±0.6** | **87.59±0.2** | **54.18±3.4** | 62.09 ± 2.2 |

Table 4: Average macro F1 scores for classification tasks on human-annotated test set over 5 runs using 5 random seeds (weighted F1 scores are shown in Appendix D.5). Entity mapping is an 11-class and the rest are 2-class prediction tasks. Author stances are determined by majority voting of the predicted stances of their tweets. Amb. means Ambiguous.

| | WEAK SUPERVISION | | DIRECT SUPERVISION | | |
| PERSPECTIVES | Our Model | Multitask | Our Model | Multitask | SUPP. |
|---|---|---|---|---|---|
| Police neg actor | **79.06±2.4** | 71.93 ± 3.7 | **81.66±1.1** | 64.02±19.0 | 255 |
| White Americans neg actor | **63.03±1.9** | 33.7 ± 2.9 | **62.76±1.6** | 19.65 ± 9.5 | 65 |
| Black Americans pos target | 72.89 ± 5.5 | **75.09±2.1** | **84.0 ± 0.9** | 83.25 ± 0.8 | 443 |
| Racism neg actor | **65.85±2.0** | 51.62 ± 5.9 | **69.07±4.1** | 60.65 ± 1.7 | 110 |
| BLM pos actor | **49.45±4.5** | 28.16 ± 6.3 | **49.52±1.3** | 47.28 ± 8.1 | 64 |
| Government neg actor | **37.58±8.8** | 29.79 ± 8.7 | **42.78±7.0** | 18.69±11.6 | 33 |
| Democrats neg actor | **62.89±5.6** | 54.06 ± 5.4 | **67.58±3.5** | 51.79 ± 5.9 | 90 |
| BLM pos target | 20.81 ± 5.5 | **28.04±5.1** | **16.36±5.9** | 0.0 ± 0.0 | 29 |
| Communities pos target | **42.87±3.2** | 42.73 ± 2.3 | **54.95±5.0** | 47.36 ± 4.2 | 61 |
| Police pos actor | **46.81±2.6** | 35.84 ± 3.6 | **45.9 ± 1.9** | 41.56 ± 1.4 | 76 |
| Police pos target | **54.17±1.7** | 40.59 ± 6.4 | **61.08±1.9** | 60.24 ± 3.1 | 138 |
| Petition pos target | 60.35±22.6 | **89.7±10.0** | **99.05±1.9** | 28.4 ± 28.3 | 10 |
| Republicans pos actor | 21.1 ± 6.4 | **29.95±5.5** | 17.4 ± 9.1 | **19.83±3.9** | 27 |
| BLM neg actor | **26.08±4.2** | 15.69 ± 7.8 | **40.0 ± 6.4** | 27.41 ± 3.5 | 76 |
| Antifa neg actor | 45.76 ± 1.7 | **46.54±1.0** | **51.56±3.6** | 43.54 ± 2.1 | 69 |
| Black Americans neg actor | **26.61±8.0** | 19.09 ± 3.4 | 5.16 ± 3.2 | 1.05 ± 2.1 | 37 |
| AVG. WEIGHTED F1 | **60.27±2.3** | 54.13 ± 1.8 | **65.93±1.4** | 56.7 ± 3.0 | 1,583 |

Table 5: F1 scores for perspective identification by our model and Multitask-RoBERTa-tapt+Author-Embed.

fails in ambiguous tweets. Next, we observe that performance improves in all tasks when all inter-dependent tasks are learned jointly in the multitask baseline compared to discrete classifiers. In multitask setup, adding author embeddings improves performance a bit in some tasks.

We study our model's performance in four settings (more details in Appendix D.5). We observe that by just converting text to graph we get an improvement over discrete text classifiers over initial text embeddings (from RoBERTa-tapt). It proves the advantage of studying texts as graphs consisting of structured elements and performing joint structured prediction on that. Next, we obtain a large gain in tweet stance classification performance when the author network is added to the text-only graph. Finally, adding the self-learning loop improves the performance a lot in all tasks and outperforms all baselines proving the effectiveness of our approach.

The trends are mostly the same in the weak supervision setup and it achieves overall comparable performance with the direct supervision setup. In this setup, adding author information does not

help in ambiguous tweet stance detection, as in this setup, the authors are not real but rather imitated and their embeddings are just the average of the generated tweets. Also, ambiguous tweet examples are not present in the LLM-generated train set.

We present the combined perspective identification results in Table 5 and observe that our model outperforms the multitask baseline in almost all perspectives. We observe low F1 scores for a few perspectives such as "Black Americans neg actor", "BLM pos target", and "Republicans pos actor". In the direct supervision setup, we find that "Black Americans neg actor" (Precision: 60%, Recall: 2.7%) and "Republicans pos actor" (Precision: 58%, Recall: 12%) get overall low average recalls. We conjecture being a less popular perspective (as shown in the support column) is the reason for low recalls. We find that 48% of the time "Republicans" wrongly map to the abstract entity "Government". In the timeline of this dataset, a Republican president was in power in the US. Hence, the framing of Republicans was probably as government figures. In the case of "BLM", 70% of the time it is mapped to a related abstract entity, "Black Americans".

Finally, we evaluate how much GPT-3 generated or human-annotated data is required for effective training of the models and find that our model is less sensitive to the number of training examples in both cases compared to the multitask baselines. The learning curves can be observed in Figure 4. The details of the ablation can be found in Appendix D.8.

### 5.3 Qualitative Evaluations

We infer perspectives and stances in the whole corpus (in Table 1) using our model and the multitask baseline and perform the following qualitative evaluations (more evaluation details can be found in Appendix E).

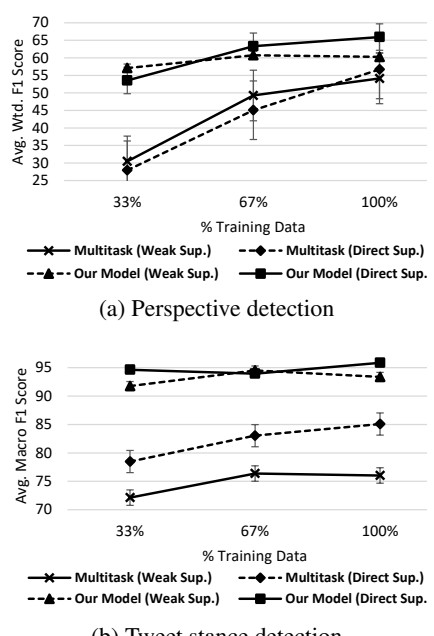

(a) Perspective detection

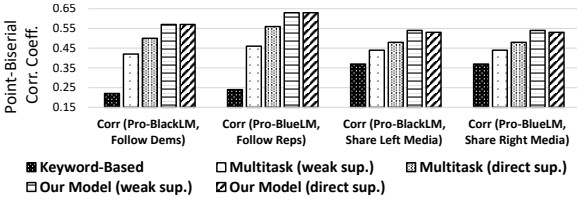

(b) Tweet stance detection

Figure 4: Learning curves for perspective and tweet stance detection for our model and the Multitask baseline.

Figure 5: Correlation between stances on #BLM movement and authors' following and sharing behaviors.

**Correlation of author behavior with stance:** We examine the correlation between authors' following and sharing behavior on Twitter with their stances on #BLM movement. We find that there are positive correlations between being pro-BlackLM and following Democrats and being pro-BlueLM and following Republicans on Twitter. We find similar correlations in sharing behavior of the authors from left and right-biased media outlets and supporting BlackLM and BlueLM, respectively. These correlations are consistent with these political camps' stances on #BLM. However, this information is not used in the learning phase, hence, it serves as an extrinsic evaluation for our model on the whole corpus. We observe that these correlations increase using our model's labels compared to baselines as shown in Figure 5, validating our models' predictions.

**Entity mapping analysis:** Depending on the author's stance, the same phrases are often used to

| Literal Entities | Most assigned perspectives by our model (direct sup.) | |
|---|---|---|
| | In pro-#BlackLM | In pro-#BlueLM |
| Black Victims | blacks pos. target | blacks neg. actor |
| Derek Chauvin | police neg. actor | N/A |
| Thugs | police neg. actor | antifa neg. actor |
| David Dorn | blacks pos. target | police pos. target |
| Lives | blacks pos. target | comm. pos. target, police pos. target |
| Donald Trump | government neg. actor | republicans pos. actor |
| They | blacks pos. target | dems. neg. actor, antifa neg. actor |

Table 6: Examples of literal entity to abstract entity map. Sometimes pro-BlackLM and pro-BlueLM use the same phrase to address different entities and/or perspectives.

| DISCOURSE IN PRO-#BLACKLIVESMATTER (inferred using our model and direct sup.) | | | |
|---|---|---|---|
| Perspectives | MFs in Context | Other Perspectives in Context | Example Tweets |
| police neg. actor | fair./cheat., auth./subv. | blacks pos. target, blm pos. actor | Say her name! #BreonnaTaylor, arrest the cops that murdered her! |
| blm movement pos. actor | loyal./betray., fair./cheat. | blacks pos. target, racism neg. actor | #Blacklivesmatter movement is exposing America society for what it really is. |
| DISCOURSE IN PRO-#BLUELIVESMATTER (inferred using our model and direct sup.) | | | |
| police pos. actor | auth./subv., loyal./betray. | police pos. target, antifa neg. actor | Protect the officers! They are only following orders and keeping America safe. |
| blm movement neg. actor | auth./subv., loyal./betray. | antifa neg. actor, democrats neg. actor | #BLM is a hateful racist organization that works to divide people... not unite. They were founded by Democrats. |

Table 7: Discourse of movements explained with messaging choices and Moral Foundations (MFs). Moral Foundation care/harm was used in all of the cases by both sides. Hence, it is removed from the table.

address different entities as shown in Table 6. Pro-BlackLM addresses police as "thugs" while pro-BlueLM addresses Antifa as "thugs". When pro-BlackLM tweets mention "lives", they mean Black lives while the pro-BlueLM means the lives of police officers. These patterns are better captured by our model compared to baselines (comparison and example tweets are shown in Appendix E.2).

**Discourse of the movements:** In Table 7, we summarize some high PMI perspectives identified in each of the movements, the corresponding moral foundations used in context of these perspectives, and the other perspectives that frequently appear in the same tweet (full list is in Appendix E.3). We infer moral foundation labels in tweets using an off-the-shelf classifier. We find that the perspectives can explain the discourse of the movements. For example, in pro-BlackLM when police are portrayed as negative actors, Black Americans are portrayed as positive targets, and BLM movement as a positive actor. Moral foundations fairness/cheating and authority/subversion are used in this context. In contrast, in pro-BlueLM, police are portrayed as positive actors and targets, and in the same context, Antifa is portrayed as a negative actor. The moral foundation of loyalty/betrayal is used in context.

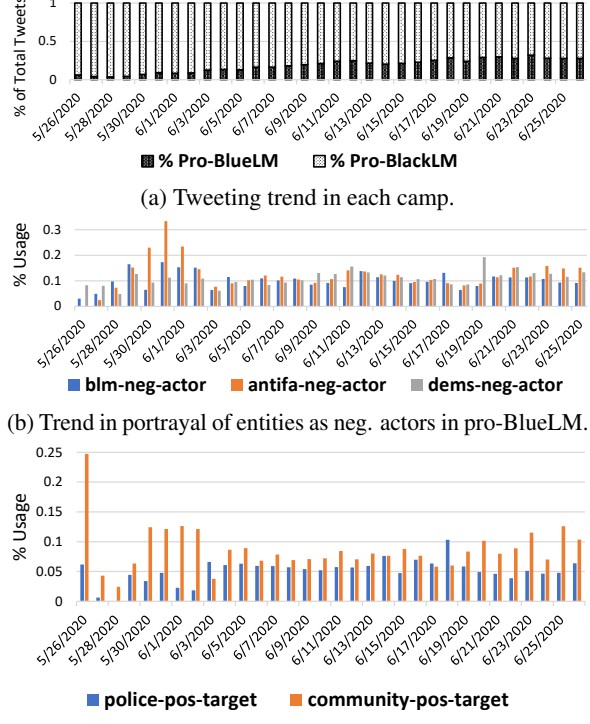

(a) Tweeting trend in each camp.

(b) Trend in portrayal of entities as neg. actors in pro-BlueLM.

(c) Trend in portrayal of entities as pos targets in pro-BlueLM.

Figure 6: Temporal trends identified using our model and direct supervision.

However, high-level moral foundations in both of the camps are sometimes similar (e.g., care/harm is frequently used with all perspectives) and entity perspectives resolve the ambiguity in those cases.

**Temporal trend:** Using our model (direct supervision), we find that $20\%$ of the tweets are identified as pro-BlueLM and the rest as pro-BlackLM. We find that first responders following George Floyd's killing were the pro-BlackLM camp. In contrast, the percentage of pro-BlueLM tweets per day slowly increased over time (Figure 6a). When the protest started (on May 26) following George Floyd's death, the pro-BlueLM camp initially portrayed BLM and Antifa as negative actors (Figure 6b) and communities to be positive targets or sufferers of the BLM movement (Figure 6c). As the movement progressed (after June 2) Democrats were portrayed as negative actors more equally. Additional trends can be found in Appendix E.4.

## 6   Related Works

The discourse of the #BLM movement is mostly studied in computational social science (CSS) using keyword-based analyses (De Choudhury et al., 2016; Gallagher et al., 2018; Blevins et al., 2019; Giorgi et al., 2022). However, it is understudied in

NLP. Early works studied the problem of identifying the names of civilians in news articles killed by police using EM-based (Keith et al., 2017) and Deep Learning approaches (Nguyen and Nguyen, 2018). Recently, Ziems and Yang, 2021 introduced a news corpus covering 7k police killings to study entity-centric framing of victims, defined as the demographics and other status (e.g., armed/unarmed). A shared task was proposed for identifying BLM-centric events from large unstructured data sources (Giorgi et al., 2021b), and Giorgi et al., 2022 introduced a large #BLM tweet corpus paving the way for more studies in this area. In this paper, we propose a holistic learning framework for understanding such social movements.

Our work is broadly related to stance detection (Küçük and Can, 2020; ALDayel and Magdy, 2021), entity-centric sentiment analysis (Deng and Wiebe, 2015; Field and Tsvetkov, 2019; Roy et al., 2021), entity disambiguation (Cucerzan, 2007; Ganea and Hofmann, 2017; Eshel et al., 2017), data augmentation (Feng et al., 2021), and the works that analyze similar discourses on social media (Demszky et al., 2019) and incorporate social supervision in language understanding such as sentiment analysis (Yang and Eisenstein, 2017), political perspective detection (Li and Goldwasser, 2019), fake-news detection (Nguyen et al., 2020; Mehta et al., 2022), and political discourse analysis (Pujari and Goldwasser, 2021; Feng et al., 2022). Detailed discussions on the CSS studies on the movements, stance, perspective analysis and data augmentation techniques can be found in Appendix F.

## 7   Conclusion

In this paper, we propose a weakly-supervised self-learned graph-based structured prediction approach for characterizing the perspectives and discourses on the #BlackLivesMatter and the #BlueLivesMatter movements on social media. We evaluate our model's performance in a human-annotated test set and find a significant improvement over all baselines. Finally, using our model we successfully analyze and compare perspectives expressed in both of the movements.

## Limitations

For the artificially crafted training data generation, we mostly use GPT-3 which is not open source and is available as a paid service[1]. Although our model

---

[1] https://beta.openai.com/playground

depends only on a few GPT-3 generated texts (cost us below $3 USD combined), generating examples at a very large scale will be expensive using GPT-3. Experimenting with the increasing number of open-source LLMs is costly in a different way as they require advanced computing resources to mount and run. Hence, we leave the study on the difference in training data generated by various LLMs as future work.

Our main focus in this paper is to develop a holistic framework that can be applied to different events related to social movements for characterizing perspectives. As a result, in this paper, we focus on one significant event related to the #BLM movement which is the outrage right after George Floyd's killing. However, our model can be applied in the case of other similar social movements that were viral on social media such as the #MeToo movement. We study two opposing movements named #BlackLivesMatter and #BlueLivesMatter in this paper. Extending this study to another relevant and potentially more ambiguous slogan, #AllLivesMatter can be interesting.

Our model depends on pre-identified abstract entities and perspectives as priors. This pre-identification of abstract entities and perspectives is a semi-automated process with human-in-the-loop or requires world knowledge. Making this step fully automatic or extracting them from an existing database can be interesting future work.

Our model does not automatically identify new abstract entities. However, in real life, new abstract entities may appear over time. This limitation does not affect our study in this paper because the study is done in a short time frame of one month and emerging of new abstract entities or change in authors' stances in this short time frame is unlikely. Extending the model to identify new abstract entities in a temporal fashion is our intended future work.

## Ethics Statement

In this section, we clarify the ethical concerns in the following aspects.

**GPT-3 Generations:** There have been concerns about inherent bias in pretrained Large Language Models (LLMs) in recent works (Brown et al., 2020; Blodgett et al., 2020). As LLMs are pretrained on large corpus of human-generated data, they may contain human bias in them. We want to clarify that, in this paper, we use LLMs to generate

biased texts that contain specific perspectives and stances. Hence, the concerns regarding different types of biases (e.g. racial, national, gender, and so on) are not applicable in the case of our study. Because we prompt LLMs to generate only a few texts that have specific structured properties (e.g., stance, sentiment towards entities), and as described in Appendix B, one author of this paper manually went through the generated examples to detect inconsistencies or any unexpected biases in the generation and no unexpected bias was observed. We believe prompting LLMs using the structured way that we propose in this paper is effective in avoiding any inconsistencies that may be additionally incorporated by the pre-trained models.

**Human Annotation:** We did human annotation of data using in-house annotators (aged over 21) and the annotators' were notified that the texts may contain sensitive phrases. The detailed annotation process and inter-annotator agreement scores are discussed in Appendix C.

**Bias and Ethics:** In this paper, we carefully addressed all communities of people and movements that appear in the dataset. We made sure that every entity is addressed with due respect. All of the sentiments, perspectives, and trends reported in this paper are outcomes of the models we developed and implemented and in no way represent the authors' or the funding agencies' opinions on this issue.

**Datasets Used:** All of the datasets used in this paper are publicly available for research and we cited them adequately.

## Acknowledgements

We gratefully acknowledge Nishanth Sridhar Nakshatri for helping with human annotation. We are thankful to Nikhil Mehta and Rajkumar Pujari for their feedback in the writing. We also thank anonymous reviewers for their insightful comments that helped improve the paper a lot. The project was partially funded by NSF CAREER award IIS-2048001.

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

## A  Identification of Abstract Entities and Corresponding Perspectives

To identify the most common high-level abstract entities or the main actors in the pro-BlackLM and pro-BlueLM tweets we follow the below steps.

- We first determine the hashtags that are mostly used in the context of the keywords "blacklivesmatter", "bluelivesmatter" and we assign soft labels of "pro-blacklivesmatter" and "pro-bluelivesmatter" to the tweets containing these hashtags, respectively. The identified "pro-blacklivesmatter" and "pro-bluelivesmatter" hashtags are shown in Table 8.

- We extract noun phrases in these tweets using SpaCy. We treat these noun phrases as entities. We remove all entities that are pronouns for this analysis.

- We build two separate RoBERTa-based (Liu et al., 2019) classifiers to identify sentiment towards entities (positive/negative) and the role of the entities (actor/target). These RoBERTa classifiers are trained using out-of-domain data. We use the annotated dataset by Roy

et al., 2021 where entities are labeled for sentiments and roles. We consider the entity labels ["target of care/harm", "target of fairness/cheating", "target of loyalty/betrayal", "target of sanctity/degradation", "authority failing over", "authority justified over"], in this dataset, as "targets" and the rest as "actors". We obtain the contextualized embeddings of entities in a tweet using RoBERTa by taking the embeddings of the last layer. Then we use a fully connected layer to identify the sentiment or role of the entity. We use $80\%$ data as training and the rest as validation set. We stop training the model when the validation accuracy does not improve for 3 consecutive epochs. We use these trained classifiers to infer the sentiment and role of entities in the #BLM corpus. The validation accuracies of both of these classifiers on out-of-domain validation sets were $> 92\%$.

- After obtaining entities, their corresponding roles, and sentiments towards them, we construct perspectives as "entity_sentiment_role", such as "police_positive_actor". Now, we obtain Pointwise Mutual Information score (PMI) (Church and Hanks, 1990) for each perspectives with the "pro-blacklivesmatter" and "pro-bluelivesmatter" stances using the following formula. For a perspective $x$ we calculate the Pointwise Mutual Information (PMI) with a stance $s$, $I(x, s)$ using the following formula.

$$I(x, s) = \log \frac{P(x|s)}{P(x)}$$

Where $P(x|s)$ is computed by taking all perspectives used in tweets with stance $s$ and computing $\frac{count(x)}{count(all-perspectives)}$ and similarly, $P(x)$ is computed by counting perspective $x$ over all of the tweets. We discard all perspectives that appear less than $0.5\%$ of the time in the whole corpus.

- We manually go through the high-PMI perspectives with each stance and cluster them to form abstract entities and perspectives. The high-PMI perspectives are shown in Table 8. We cluster entities and perspectives that are consistent and directed to the same set of entities. For example, in pro-bluelivesmatter tweets, "a riot_neg_actor" and "# blacklivesmatter # protests_neg_actor" are directed to

the same high-level entity "BLM Movement" and express the same perspectives. Hence, we merge these entities to the abstract entity "BLM Movement" and identify the relevant perspective towards this entity in probluelivesmatter as "Negative Actor". The descriptions of all identified abstract entities can be found in Table 9.

## B Prompting LLMs to Generate Few Training Tweets

In this section, we describe the details of few training data generation using Large Language Models (LLMs). We conduct our initial experiments with two LLMs - GPT-3 (175B parameters) (Brown et al., 2020) and GPT-J (6B parameters) (Wang and Komatsuzaki, 2021) and find that GPT-J generates a lot of repetitive examples. We conjecture GPT-J being a very small LLM in terms of parameters is the main reason for a less diverse generation. This is in line with recent findings where larger LLMs were found to perform better in various tasks than the smaller LLMs. Hence, we retain only the GPT-3 generations in this paper. The generation process using GPT-3 is described below.

### B.1 Generation

For generating a few tweets containing the stances and corresponding entity perspectives (as described in Table 2), we prompt the GPT-3 model (Brown et al., 2020) using the prompt structure shown in Figure 7. We prompt GPT-3 in such a way that all of the structured elements for a stance and corresponding perspective are present in the generated tweets. To ensure that, as shown in Figure 7, we instruct GPT-3 model to generate 30 different tweets that must fulfill the following three conditions.

(1) Explicitly mention one or more entities from the following list *<entity list>* and portray them as *<entity perspective>*.

(2) Be in the support of the *<movement name>* movement.

(3) Use one or more hashtags from the following list *<hashtag list>*.

Here, *<entity perspective>* can be one of the following - "positive target", "positive actor", "negative actor" and *<movement name>* is either "#BlackLivesMatter" or "#BlueLivesMatter".

We take the *<entity list>* and *<hashtag list>* from Table 8. We perform a separate prompting step for each (stance, abstract entity, entity perspective) tuple in Table 2. For example, to generate tweets that are "pro-BlackLM" and portray "Black Americans" as "positive target", we prompt GPT-3 by using the prompt shown in Figure 3. We find that GPT-3 rarely generates repetitive tweets. We prompt GPT-3 multiple times for generating at least 20 unique examples per (stance, abstract entity, entity perspective) tuple. One author of this paper skims through the GPT-3 generated examples to detect any inconsistency or unwanted bias in the generations and discards the generation if found any. The observation is that the GPT-3 generations are mostly clean. We use OpenAI interface[2] for generating the tweets using *text-davinci-003* (largest available) version of GPT-3 (till January 2023) using the following hyperparameters: top-p=1, frequency penalty: 1, presence penalty=1, temperature=1, max len=500. Note that, on the OpenAI console usage of GPT-3 for any experiment is a paid service. All of the generations used in this paper cost us in total $\sim$ \$3 US Dollars.

### B.2 Preprocessing and Labeling of Generated Tweets

After the generation of tweets using GPT-3 that contain the elements from (stance, abstract entity, entity perspective) tuples, we preprocess these tweets to identify the text span containing the abstract entities. Note that, an abstract entity can be addressed differently in different tweets. For example, "police" can be addressed as "cops" or "law enforcement". To identify the abstract entity containing spans in generated tweets, we first run SpaCy noun phrase extractor on these tweets and automatically group the extracted entities based on keyword match, for example, "police" and "police force" will be grouped together because of the common keyword "police". Then an author of this paper looks at the entity groups and discards entity groups that are not related to the target abstract entity. Then we consider the rest of the entity groups and annotate them as gold data for abstract entity mapping. The statistics of generated data are shown in Table 10. Two examples of GPT-3 generated tweets for each perspective are shown in Table 11. We submit a randomly selected subset (20 tweets) of the GPT-3 generated annotated data with this manuscript for review. Upon acceptance of this paper, we will release the whole set.

---

[2]https://beta.openai.com/playground

| Stances | Top 20 Most used hashtags | High PMI perspectives |
|---------|---------------------------|------------------------|
| **Pro Blacklivesmatter** | #blacklivesmatter, #blm, #georgefloyd, #alllivesmatter, #policebrutality, #justiceforgeorgefloyd, #defundthepolice, #racism, #nojusticenopeace, #covid19, #icantbreathe, #breonnataylor, #bluelivesmatter, #protests2020, #antifa, #trump, #blacklivesmatteruk, #policebrutalitypandemic, #protests, #georgefloydprotests | 'the activists_pos_target', '# ahmaudarbery_pos_target', '# breonnataylor_pos_target', 'a petition_pos_target', '# justice_pos_actor', '# dcprotests_pos_target', 'poc_pos_target', '# justice_pos_target', '# georgefloyd_pos_target', '# blacklivesmatter # protests_pos_target', 'a black man_pos_target', 'freedom_pos_target', 'a white man_pos_target', '# america_pos_actor', '# rayshardbrooks_pos_target', '# blacklivesmatter # protests_pos_actor', 'african americans_pos_target', 'the kkk_neg_actor', 'government_neg_actor', '# whiteprivilege_neg_actor', '# whitesupremacy_neg_actor', 'a white man_neg_actor', '# racism_neg_actor', '# joebiden_neg_actor', '# trump_neg_actor', 'a black man_neg_actor', '# america_neg_actor', '# policeviolence_neg_actor', 'republicans_neg_actor', 'media_neg_actor', 'president_neg_actor' |
| **Pro Bluelivesmatter** | #bluelivesmatter, #alllivesmatter, #blacklivesmatter, #backtheblue, #whitelivesmatter, #maga, #trump2020, #bluelivesmatters, #maga2020, #blm, #thinblueline, #whiteoutwednesday, #womenfortrump, #kag, #buildthewall, #alllivesmatters, #blueflu, #police, #defundthepolice, #lawenforcement | '# cops_pos_target', '# cops_pos_actor', 'law_pos_target', 'republicans_pos_target', 'god_pos_actor', 'communities_pos_target', '# daviddorn_pos_target', 'the country_pos_target', '# equality_pos_target', '# america_pos_target', 'a riot_neg_actor', '# democrats_neg_actor', '# blacklivesmatter # protests_neg_actor', 'politicians_neg_actor', '# antifa_neg_actor', '# cops_neg_actor', '# obama_neg_actor', 'looters_neg_actor' |

Table 8: Data analysis for abstract entity and corresponding perspectives identification.

| Abstract Entities | Description |
|-------------------|-------------|
| **Black Americans** | Refers to Black Americans. |
| **Police** | Refers to the Police department and law enforcement in the USA. |
| **Community** | Refers to the entities that refer to communal spirit such as communities, society, nation, United States of America, citizens, etc. |
| **Racism** | Refers to racism and racists. |
| **Democrats** | Refers to the Democratic party in the USA, politicians from this party, or anyone supporting them. |
| **Republicans** | Refers to the Republican party in the USA, politicians from this party, or anyone supporting them. |
| **Government** | Refers to the government or any authoritative figures such as president, governors, mayors, and so on. |
| **White Americans** | Refers to White Americans. |
| **BLM Movement** | Refers to the BlackLivesMatter movement, the protesters, the activists, and the supporters of this movement. |
| **Petition** | Refers to any official petition or campaign for supporting a cause. |
| **Antifa** | The Anti-Fascist and Anti-Racist political movement in the USA. |

Table 9: Abstract entities and their descriptions.

## C   Real Data Annotation Procedure

We human-annotate a subset of the whole #BLM corpus as described in Table 1 for the following two purposes.

- To evaluate the performance of our proposed model on real data and compare it with the baselines.

- To evaluate the quality of the artificially crafted training data (using LLMs) by comparing the performance of our proposed model

and the baselines that are trained with the artificially crafted data vs. the real annotated data.

The human annotation of real data occurs in the following three steps.

**Step-1:** From the whole #BLM corpus summarized in Table 1, we rank authors who are most consistent in the usage of the keywords blacklivesmatter and bluelivesmatter. The most consistent user uses only either of the keywords 100% of the time. Then we randomly sample 100 users from the top 500 consistent users of blacklivesmatter and bluelivesmatter resulting in 200 users. We annotate these users for their stance (pro-blacklivesmatter or pro-bluelivesmatter). We present two human evaluators with each author's id, their profile description, and all of the tweets they shared. Note that the author names were hidden and the author ids were just numeric ids assigned by Twitter to Twitter users. Then we ask the human annotators to annotate the authors for stance by looking at their profile descriptions and the tweets they shared. The human annotators are asked to annotate 'none' if it is not possible to infer the stance of an author by looking at the tweets and the profile description. We find an average inter-annotator agreement of 0.824 (almost perfect agreement) using Cohen's Kappa measure.

```
<prompt-start>
Write 30 different tweets that must -
1. Explicitly mention one or more entities from the following list <entity list> and portray
them as <entity perspective>.
2. Be in the support of the <movement name> movement.
3. Use one or more hashtags from the following list: <hashtag list>.

Answer:
<prompt-end>

<generation-start>
1. <generated tweet>
2. <generated tweet>
3. <generated tweet>
......
......
```

Figure 7: Prompt structure for GPT-3 to generate tweets containing specific stances and perspectives. The black-colored text is the input text in the prompt and the blue-colored texts are generated tweets by GPT-3.

| Stance: Perspective | # Generated tweets | # Generated tweets having entity mention | # Entities |
|---|---|---|---|
| pro-blacklivesmatter: whites neg actor | 30 | 27 | 41 |
| pro-blacklivesmatter: blm movement pos target | 30 | 28 | 28 |
| pro-blacklivesmatter: petition pos target | 30 | 30 | 36 |
| pro-blacklivesmatter: police neg actor | 50 | 33 | 31 |
| pro-blacklivesmatter: blm movement pos actor | 30 | 30 | 30 |
| pro-blacklivesmatter: government neg actor | 30 | 28 | 35 |
| pro-blacklivesmatter: blacks pos target | 40 | 30 | 30 |
| pro-blacklivesmatter: racism neg actor | 29 | 24 | 27 |
| pro-bluelivesmatter: community pos target | 44 | 28 | 34 |
| pro-bluelivesmatter: police pos actor | 36 | 34 | 28 |
| pro-bluelivesmatter: antifa neg actor | 29 | 27 | 30 |
| pro-bluelivesmatter: police pos target | 46 | 33 | 30 |
| pro-bluelivesmatter: blm movemen neg actor | 30 | 30 | 24 |
| pro-bluelivesmatter: republicans pos actor | 58 | 28 | 47 |
| pro-bluelivesmatter: blacks neg actor | 40 | 40 | 40 |
| pro-bluelivesmatter: democrats neg actor | 30 | 26 | 26 |
| **Total** | 582 | 476 | 517 |

Table 10: GPT-3 generated training data statistics.

The disagreements between the two annotators are resolved by discussion. After annotating authors for stances, we readily get stance labels for the tweets they wrote. We find that often the supporters of the movements use the keywords/hashtags that are used mostly by the counter-movement to troll or criticize the opponent. This is in line with previous findings in computational social science (Gallagher et al., 2018). We annotate these tweets as "Ambiguous Tweets". Ambiguous tweets are intuitively more difficult for models to disambiguate because they use the keywords and hashtags that are frequently used by the opponents. We determine a tweet to be an ambiguous tweet if it is annotated by human annotators as "pro-blacklivesmatter" but

the tweet uses the keyword "bluelivesmatter" and vice versa for "pro-bluelivesmatter". Examples of ambiguous tweets are presented in Table 12.

**Step-2:** After annotating tweets and authors for stances we extract entities from these tweets using SpaCy noun phrase extractor and in this step, we annotate these entities for abstract entity labels. We present two human annotators with a tweet, its stance (annotated in the previous step), and an entity mentioned in the tweet. Then we ask the annotators to assign an abstract entity label to the entity from the list of abstract entities in Table 2. The annotators could select multiple abstract entity labels or "none" for an entity. We find an average Cohen's Kappa inter-annotator agreement score of

| Stance: Perspective | GPT-3 Generated Tweets |
|---|---|
| **pro-blacklivesmatter: blacks pos target** | 1. We must continue to fight for justice and equality for all black people. #alllivesmatter #police-brutalitypandemic
2. We will no longer tolerate the unjust murder of black people by those in positions of power. #blm #policebrutalitypandemic |
| **pro-blacklivesmatter: blm movement pos actor** | 1. The Black Lives Matter protests have been effective in bringing awareness to the police brutality pandemic. #blacklivesmatter #blm
2. The Black Lives Matter movement is helping to raise awareness about the issues faced by minorities. #blacklivesmatter |
| **pro-blacklivesmatter: blm movement pos target** | 1. I stand with the #BlackLivesMatter movement and protest against police brutality! #policebrutality #blm
2. Join the fight to end police brutality and racism by supporting the #BlackLivesMatter movement! We can make a difference by coming together and speaking out against these injustices! |
| **pro-blacklivesmatter: government neg actor** | 1. From Ferguson to Baltimore, it's clear that the government doesn't care about us. Time for change! #blacklivesmatter
2. The government has blood on their hands and we won't rest until they're gone! #blacklivesmatter #blm |
| **pro-blacklivesmatter: petition pos target** | 1. Sign the petition to hold police accountable for their brutality! #blacklivesmatter
2. Show your support and sign the petition now! #alllivesmatter |
| **pro-blacklivesmatter: police neg actor** | 1. The police are supposed to protect us, not kill us. How many more innocent lives will be lost before something is done? #policebrutalitypandemic
2. Police brutality must end now! Outrage is not enough, we demand action! #PoliceBrutalityPandemic |
| **pro-blacklivesmatter: racism neg actor** | 1. Racism, like police brutality, is a systemic problem that needs to be addressed immediately. #blacklivesmatter #blm
2. Silence is complicity! If you're not part of the solution, you're part of the problem. Show your support and Stand up against bigotry! #blacklivesmatter |
| **pro-blacklivesmatter: whites neg actor** | 1. #whiteprivilege is real and it's something that needs to be acknowledged and dismantled. No one should have power over another because of the color of their skin
2. It's time to take a stand against the hate and bigotry of #whitesupremacists. We must come together and show that we are committed to equality and justice for all. |
| **pro-bluelivesmatter: antifa neg actor** | 1. Antifa domestic terrorists are ruining our country! #bluelivesmatter #alllivesmatter
2. I stand with the brave men and women of law enforcement who keep our country safe from groups like antifa! #backtheblue |
| **pro-bluelivesmatter: blacks neg actor** | 1. It's always the black people who cause the most trouble in our community. #alllivesmatter #bluelivesmatter #backtheblue
2. We can't trust black people to obey the law. #whitelivesmatter #alllivesmatter #backtheblue |
| **pro-bluelivesmatter: blm movement neg actor** | 1. The blacklivesmatter protests are nothing but an excuse to riot and loot! #bluelivesmatter #alllivesmatter
2. BLM protests put innocent lives at risk! #bluelivesmatter |
| **pro-bluelivesmatter: community pos target** | 1. We stand with our communities and the police who keep us safe! #bluelivesmatter #backtheblue
2. I'll never understand why people want to harm police officers who are just trying to do their job and keep our communities safe. #BlueLivesMatter |
| **pro-bluelivesmatter: democrats neg actor** | 1. The left is always championing criminals and trying to tear down our police officers. #bluelivesmatter #backtheblue
2. Dems don't care about law and order, they only care about chaos and anarchy. #bluelivesmatter |
| **pro-bluelivesmatter: police pos actor** | 1. Police put their lives on the line every day to protect us! #blacklivesmatter
2. The media portrays law enforcement in a negative light but I know they are doing an amazing job!!! #ProudOfOurPolice |
| **pro-bluelivesmatter: police pos target** | 1. Thank you to all the brave men and women in law enforcement who keep us safe! #bluelivesmatter #backtheblue
2. I stand with the police and oppose the violence against them! #AllLivesMatter |
| **pro-bluelivesmatter: republicans pos actor** | 1. We are proud conservatives for standing behind all police officers as they work to make our communities safer - let's keep on building an America where blue lives matter! #whitelivesmatter
2. The Republican party stands firmly behind those who serve and protect us each day – it is essential to acknowledge their efforts and reject any attempts to discredit them. #bluelivematters |

Table 11: Examples of GPT-3 generated tweets.

0.697 (substantial agreement) for this task. The disagreements are resolved by discussion. We find that 78% of the entities map to at least one abstract entity and a small portion of entities (<5%) are identified to map to multiple abstract entities. It implies that the abstract entities summarized in Table 2 are the main actors related to the movements and they have very good coverage of the whole data.

Based on the feedback from the human annotators, the unmapped entities are mostly cases where it is difficult to determine what abstract entity they are referring to without more context and a small portion of them were wrong entity detection by the SpaCy noun phrase extractor.

**Step-3:** In this step, we annotate the entities annotated for abstract entity labels in Step-2, for

| Ambiguous Tweets | Annotated Stance |
|---|---|
| SPAMMING RACIST TAGS #Qaṇöṇ #trump2020 #trump #EXPOSEANTIFA #ExposeAntifaTerrorists #bluelivesmatter #Whitelivesmatter #whitelivesmattermost #whitelivesmattermore #whitelivesmattertoo #WWG1WGA | pro-blacklivesmatter |
| Of course #BlueLivesMatter mailmen are the backbone of the nation | pro-blacklivesmatter |
| Tell me about the good cops #BLUEFALL #BlueLivesMatter | pro-blacklivesmatter |
| #BlackLivesMatters protestors are racist cop killers and vandals Not FoxNews @Twitter should remove the lying hashtag #FoxNewsisRacist as these thugs vandalize memorials of war heros on #DDay | pro-bluelivesmatter |
| #BLM Burn. Loot. Murder. #BlackLivesMatter is a joke. | pro-bluelivesmatter |
| #BlackLivesMatter ... They really don't, to the black lives group! They must have gotten their group minutes from #Metoomovement #LIARS | pro-bluelivesmatter |

Table 12: Examples of tweets and their human-annotated stances where supporters of a movement use hashtags/keywords related to the counter-movement to criticize or troll them. We annotate these tweets as ambiguous tweets.

sentiments toward them (positive/negative) and assigned roles (actor/target). We present two human annotators with the tweet text, its stance, an entity mentioned in the tweet, and the abstract entity label of the entity (all determined in the previous two steps). Then we ask them to annotate the entity for role and sentiment. For role identification, the annotators are instructed to select "none" if it is not clear from the tweet text what the role of the entity is or select "both" if the entity is portrayed both as an actor and a target. For example, in the tweet "Police keep us safe. We should defend our Police.", the entity "Police" is portrayed to be both a positive actor and a positive target. We find Cohen's Kappa inter-annotator agreement scores of $0.976$ and $0.815$ for entity sentiment and entity role annotation, respectively, that are almost perfect. We resolve the disagreements by discussion. In $< 1\%$ cases, we find an entity to have both the actor and target roles in a tweet.

The per-label agreement scores for each annotation task can be found in Table 13 and the final annotated data statistics can be found in Tables 14 and 15.

Both of the human annotators were graduate students (age above 21) and they were awarded research credits for this annotation task. They were sufficiently briefed on the tasks and were informed that the tweets may contain potentially sensitive language. The annotators were also informed that the dataset will be used for research purposes.

**Data Consolidation:** As described above, for the entity role annotation we find $< 1\%$ entities that map to multiple labels. We discard these entities

from the annotated dataset as all the models we trained are trained to predict one class and intuitively these cases are difficult for even humans to disambiguate. In the entity mapping annotation, we find $< 5\%$ entities that map to multiple abstract entities. We randomly select one abstract entity from the multiple abstract entity labels as the final label. It results in $189$ users annotated for stance, $2,980$ tweets annotated for stance ($520$ among them are annotated as ambiguous tweets), and $2,091$ entities annotated for abstract entity labels, sentiment toward them, and their roles.

**Human-Annotated Training and Test Data Selection:** We randomly sample 50 authors from the 189 annotated authors. These 50 authors, their tweets, and the entities mentioned in those tweets are defined as human-annotated real training data. We define the rest of the annotated dataset as human-annotated test data. Our proposed model and the baselines are trained in the weakly supervised setting using the GPT3-generated training data and in the directly supervised setting, they are trained using the human-annotated real data. In both cases, the models are tested on the human-annotated test dataset. The statistics of the GPT3-generated training set, the human-annotated training set, and the human-annotated test set are shown in Table 3.

## D    Experimental Setting

### D.1    Preprocessing of data

We collect the tweet texts of the tweet ids provided in the source dataset (Giorgi et al., 2022) using

**AUTHOR STANCE ANNOTATION**

| Stance | Agreement |
|---|---|
| Pro #BlackLivesMatter | 0.824 |
| Pro #BlueLivesMatter | 0.824 |
| **Average** | **0.824** |

**ENTITY MAPPING ANNOTATION**

| Abstract Entities | Agreement |
|---|---|
| antifa | 0.530 |
| blacks | 0.834 |
| blm movement | 0.728 |
| community | 0.442 |
| democrats | 0.706 |
| government | 0.623 |
| petition | 1.0 |
| police | 0.913 |
| racism/racists | 0.553 |
| republicans | 0.689 |
| whites | 0.653 |
| **Average** | **0.697** |

**ENTITY SENTIMENT ANNOTATION**

| Sentiment | Agreement |
|---|---|
| Positive | 0.976 |
| Negative | 0.976 |
| **Average** | **0.976** |

**ENTITY ROLE ANNOTATION**

| Role | Agreement |
|---|---|
| Actor | 0.782 |
| Target | 0.847 |
| **Average** | **0.815** |

Table 13: Inter annotators agreement for the data annotation process. Cohen's Kappa scores are used as agreement.

| Abs. Entities | Count | Sentiment | | Role | |
|---|---|---|---|---|---|
| | | Pos | Neg | Actor | Target |
| police | 784 | 464 | 320 | 439 | 280 |
| whites | 76 | 6 | 70 | 72 | 3 |
| black-people | 583 | 539 | 44 | 61 | 516 |
| racism/racists | 130 | 0 | 130 | 130 | 0 |
| blm movement | 286 | 198 | 88 | 158 | 30 |
| democrats | 111 | 7 | 104 | 111 | 0 |
| government | 46 | 3 | 43 | 46 | 0 |
| republicans | 79 | 45 | 34 | 66 | 6 |
| communities | 71 | 70 | 1 | 1 | 70 |
| petition | 11 | 11 | 0 | 0 | 11 |
| antifa | 91 | 7 | 84 | 91 | 0 |
| **Total** | **2268** | **2268** | | **2091** | |

Table 14: Annotated data statistics for entities.

Twarc API calls[3]. Before using the #BLM corpus collected from (Giorgi et al., 2022), we remove all non-ASCII characters and URLs from the tweet text. We used SpaCy Noun Phrase Extractor to

---

[3] https://twarc-project.readthedocs.io/en/latest/api/client/

| | Pro #BlackLM | Pro #BlueLM | Total |
|---|---|---|---|
| Authors | 122 | 67 | 189 |
| All Tweets | 1943 | 1037 | 2980 |
| Ambiguous Tweets | 314 | 206 | 520 |

Table 15: Annotated data statistics for authors and tweets.

extract the entities from tweet text.

To extract keywords from the author profile descriptions, first, we identify all hashtags used in the description and add them to the keywords list, then we extract ngrams ($1 \leq n \leq 3$) from the residual text. Then we merge each ngram to a single word and check if it is similar to a hashtag (e.g., merged ngram "Black lives matter" is similar to "#blacklivesmatter"). If it is similar we add that ngram to the keyword list. For the residual ngrams we look at the most occurring ones and manually discard those that do not imply any meaningful message. We add the rest of the ngrams to the keywords list.

Each element summarized in Table 1, corresponds to a unique node in our graph. For example, if two tweets mention lexically equal entities, the entities will have two different node representations in the graph. Because the perspectives towards the lexically equal entities may be different in the two tweets. For example, "law enforcement" may be portrayed as "positive actor" in one tweet and "negative actor" in another. Hence, the number of nodes and edges in our graph corresponds to the statistics in Table 1.

### D.2 Task Adaptive Pretraining of RoBERTa

Following previous works (Gururangan et al., 2020), we perform task adaptive pretraining of RoBERTa for our task. We continue pretraining RoBERTa with the whole word masking technique. In this approach, we randomly select some words and mask the whole word. Then we predict the original vocabulary ID of the masked word based on the context it appears in. We use our unused data of the #BLM corpus from (Giorgi et al., 2022) in this pretraining step. We find that this task adaptive pretraining improves classification results significantly over simple RoBERTa as shown in Tables 4 and 16.

| | MODELS | AUTHOR STANCE | | ALL TWEET STANCE | | AMB. TWEET STANCE | | ENTITY SENTIMENT | | ENTITY ROLE | | ENTITY MAPPING | |
|---|---|---|---|---|---|---|---|---|---|---|---|---|---|
| | | Weak Sup. | Direct Sup. | Weak Sup. | Direct Sup. | Weak Sup. | Direct Sup. | Weak Sup. | Direct Sup. | Weak Sup. | Direct Sup. | Weak Sup. | Direct Sup. |
| **NAIVE** | Random | 53.71 ± 1.87 | | 51.55 ± 0.45 | | 49.47 ± 1.85 | | 50.19 ± 1.61 | | 50.18 ± 1.37 | | 10.3 ± 0.7 | |
| | Keyword Based | 89.75 ± 0.85 | | 89.19 ± 0.31 | | 22.68 ± 1.4 | | - | | - | | - | |
| **DISCRETE** | RoBERTa | 70.83±11.3 | 81.0 ± 5.3 | 67.38 ± 7.3 | 77.02 ± 2.5 | 30.24 ± 3.3 | 53.31 ± 5.0 | 76.5 ± 1.6 | 80.71 ± 1.3 | 75.82 ± 0.5 | 82.92 ± 1.6 | 55.99 ± 3.0 | 67.93 ± 2.8 |
| | RoBERTa-tapt | 78.17±12.7 | 87.86 ± 2.2 | 76.31±10.6 | 84.74 ± 1.6 | 34.25 ± 1.8 | 67.81 ± 6.4 | 84.43 ± 1.4 | 86.27 ± 0.2 | 84.96 ± 1.0 | 86.55 ± 0.6 | 59.99 ± 4.4 | 64.46 ± 1.2 |
| **MULTITASK** | RoBERTa | 75.77 ± 6.0 | 84.19 ± 4.2 | 68.14 ± 6.1 | 80.55 ± 3.4 | 31.18 ± 1.6 | 54.71 ± 4.8 | 77.15 ± 0.8 | 79.37 ± 0.7 | 75.59 ± 1.5 | 84.17 ± 0.9 | 62.92 ± 1.4 | 62.34 ± 4.2 |
| | RoBERTa-tapt | 80.51 ± 4.1 | 91.3 ± 1.5 | 77.72 ± 5.0 | 87.8 ± 1.1 | 35.01 ± 1.3 | 73.03 ± 2.1 | 84.89 ± 1.1 | 86.39 ± 0.3 | 84.23 ± 0.6 | 87.69 ± 0.5 | 62.55 ± 3.5 | 63.97 ± 1.7 |
| | + Author Embed. | 82.82 ± 1.4 | 91.46 ± 1.8 | 77.2 ± 2.3 | 86.74 ± 0.9 | 34.24 ± 0.8 | 71.7 ± 4.9 | 85.23 ± 1.1 | 86.89 ± 0.4 | 83.58 ± 0.7 | 87.52 ± 0.2 | 60.05 ± 2.5 | 64.96 ± 2.1 |
| **OUR MODEL** | Text-discrete | 72.47 ± 9.7 | 80.86 ± 3.7 | 70.02 ± 8.7 | 68.67 ± 4.7 | 34.38 ± 2.0 | 62.46 ± 4.7 | 83.37 ± 0.9 | 83.84 ± 0.5 | 84.44 ± 0.9 | 83.54 ± 0.9 | 49.93 ± 2.9 | 25.04±15.3 |
| | Text-as-Graph | 80.56 ± 3.4 | 80.78 ± 1.7 | 79.76 ± 3.1 | 81.16 ± 1.4 | 36.63 ± 4.9 | 43.26 ± 2.2 | 84.96 ± 0.2 | 85.37 ± 0.5 | 86.36 ± 0.1 | 86.27 ± 0.4 | 61.71 ± 2.1 | 72.86 ± 0.5 |
| | + Author Network | 83.51 ± 2.2 | 94.0 ± 1.0 | 90.6 ± 1.5 | 96.2 ± 0.7 | 38.14 ± 0.1 | 87.28 ± 4.0 | 85.02 ± 0.2 | 84.74 ± 0.5 | 86.46 ± 0.1 | 86.26 ± 0.3 | 62.56 ± 3.2 | **73.47±1.6** |
| | + Self-Learning | **91.76±1.1** | **95.94±1.6** | **94.1 ± 0.4** | **96.33±2.5** | **64.36±5.1** | **92.48±4.7** | **87.1 ± 0.5** | **87.72±0.4** | **87.14±0.6** | **87.95±0.2** | **67.53±2.4** | 73.34 ± 1.5 |

Table 16: Average weighted F1 scores with standard deviations for classification tasks over 5 runs using 5 random seeds: 1000, 2000, 3000, 4000, 5000.

## D.3 Our Model Initialization and Hyperparameters

We initialize the representation of each node type in our graph using this RoBERTa-tapt. Author nodes are initialized by RoBERTa-tapt embeddings of the author profile descriptions or average embedding of randomly sampled 5 tweets by the author if no profile description is found.

We pretrain the external classifiers ($C_{sent}$, $C_{role}$) on out-of-domain data proposed by Roy et al. (2021). In this dataset, entities mentioned in tweets from US politicians are annotated for their moral roles. The moral roles are associated with positive and negative sentiments and actor and target types. As a result, we get annotated dataset for pretraining our external classifiers, $C_{sent}$ and $C_{role}$, for sentiment and role classifications, respectively. To train these classifiers, contextualized embeddings of the entities are obtained using RoBERTa-tapt, and a fully-connected layer is used to identify role/sentiment in these two classifiers. We train these classifiers until the accuracy in a held-out validation set (20% from OOD) do not improve for three consecutive epochs. The accuracies of both of these classifiers on the out-of-domain validation sets were $> 92\%$. We stop backpropagating to RoBERTa-tapt when $C_{sent}$ and $C_{role}$ are combined with our framework, however, the fully-connected layers are updated.

We use a 2-layer R-GCN to encode our graph nodes and $768d$ input node features are learned in $100d$ and $50d$ spaces in the 2 layers of the R-GCN. We use a learning rate of $0.0005$ to train the model. We infer after every 10 steps and before the first inference step, we train the model until the total training loss does not increase for 3 consecutive epochs. We stop training the model if the number of new training examples is less than $0.3\%$ for 10 consecutive inference steps or a maximum epoch of 300 is reached. For consistency check, we use a label confidence threshold of 0.9 till epoch 200 and reduce it to 0.8 after that, as many training examples are already added till then and the model is intuitively more stable. We set the tweet threshold, t for author consistency check to 10, 5, and 3 at epochs 1, 20, and 50, respectively. We find that the majority number of tweets ($> 75\%$) become consistent and are added to the self-learned training set till epoch 300. The hyperparameters are determined empirically.

## D.4 Baselines

**Naive:** We implement two naive baselines. The first one is a random selection where labels of tweet stance, entity sentiment, entity role, and entity mapping are determined by a random selection of corresponding labels. We perform the random selection using five different random seeds and report the average results. The second naive baseline is applicable for only tweet stance classification. We follow a keyword-matching approach similar to Giorgi et al., 2021a for that. For the classification of tweet stance using keyword-matching, we use the keywords available with each data point in (Giorgi et al., 2021a). In this dataset, each tweet is marked with one or more of the following three keywords [blacklivesmatter, bluelivesmatter, alllivesmatter] based on their presence in the tweet. We classify a tweet to be pro-BlackLM if it is marked to have the keyword blacklivesmatter in it and label it as pro-BlueLM if it contains bluelivesmatter. In case of a tie or not availability of any one of these keywords, we break it randomly. We do not consider the keyword alllivesmatter in this classification approach because this keyword is more ambiguous and used widely by both of the movements. We determine the author stances based on the majority voting on the identified stances of the tweets they wrote.

**Discrete Text Classifiers:** We implement the second type of baseline that are discrete text classifiers

based on pre-trained RoBERTa. We finetune separate RoBERTa-based classifiers for each of the tweet stance, entity sentiment, entity role, and entity mapping classification tasks.

For the classification of tweet stances, we encode the tweet text using RoBERTa-tapt. The representation of the [CLS] token of the last hidden layer is used as the tweet embedding and it is passed through a fully connected layer to predict the stance of the tweet. We update RoBERTa-tapt parameters as well during the learning steps. We stop learning when the validation accuracy does not improve for three consecutive epochs. The author stances are determined based on the majority voting on the identified stances of the tweets they wrote.

For the classification of entity roles, sentiments, and mapping we encode the entity-mentioning segments in the tweet texts using RoBERTa-tapt. We take the representation of the last layer and select tokens corresponding to the entity spans in the tweet text. Then we average these selected tokens' embeddings to get the representation of the entity. Then a fully connected layer is used to predict either of sentiment, role, or abstract entity mapping of the target entity. We update RoBERTa-tapt parameters as well during the learning steps. We stop learning when the validation accuracy does not improve for three consecutive epochs. To make the entity role and sentiment classification baselines comparable to our proposed model, during training, we combine the OOD data from (Roy et al., 2021) with the LLM-generated training data or the human-annotated training data in the weak and direct supervision settings, respectively.

**Multitask:** Our model jointly models perspectives with respect to entities (sentiment towards them, assigned role, abstract entity mapping) and the stances in the tweets. Context-rich representations of entities and tweets are learned and the single unified representation is used to infer various labels such as stance for tweets and, sentiment, role, and mapping for entities. As a result, the closest match to our model is the multitask approach.

To implement the multitask baseline, we define a single pre-trained RoBERTa-tapt text encoder that is shared across all classification tasks such as tweet stance, entity sentiment, entity role, and entity mapping.

For the classification of tweet stances, we encode the tweet text using the shared RoBERTa-tapt encoder. The representation of the [CLS] token of the last hidden layer is used as the tweet embedding and it is passed through a task-specific two-hidden-layer feed-forward neural network to predict the stance of the tweet.

For the classification of entity roles, sentiments, and mapping, we encode the entity-mentioning segments in the tweet texts using the shared RoBERTa-tapt encoder. We take the representation of the last layer and select tokens corresponding to the entity spans in the tweet text. Then we average these selected tokens' embeddings to get the representation of the entity. Then three different task-specific two-hidden-layer feed-forward neural networks are used to predict the sentiment, role, and abstract entity mapping of the target entity, respectively.

We define the multitask loss function, $L_M$ as follows.

$$L_M = \lambda_1 L_{stance} + \lambda_2 L_{sent} + \lambda_3 L_{role} + \lambda_4 L_{map}$$

Here, $L_{stance}$ is tweet stance classification loss, and $L_{sent}$, $L_{role}$ and $L_{map}$ are entity sentiment, role, and mapping classification losses, respectively. All of the losses are Cross Entropy losses and we set, $\lambda_1 = \lambda_2 = \lambda_3 = \lambda_4 = 1$ for our experiments.

We pretrain the shared RoBERTa-tapt and the sentiment and role classification-specific neural networks with the OOD data from (Roy et al., 2021) in the same way we pretrain the $C_{sent}$ and $C_{role}$ classifiers that are used in our model. All the task-specific feed-forward neural networks use two hidden layers consisting of 300 and 100 neurons with ReLU activations. We keep training the shared RoBERTa-tapt and the task-specific classifiers using either the artificial training data or human-annotated real training data. We stop training when the combined tweet stance, entity sentiment, role, and mapping classification accuracy do not improve for five consecutive epochs.

**Multitak with Author Information:** The multitask approach described above jointly models only textual features. However, our proposed model (as described in Section 3), is able to incorporate author information and social interaction among them. Hence, for a fair comparison of the multitask baseline with our proposed model, we enhance the multitask baseline with author information.

First of all, we learn rich author embeddings using relational graph convolutional networks (R-GCN). For that, we create a graph consisting of

only author nodes and keyword nodes (as described in Section 3). Two authors are connected to each other with the retweet relationships and an author is connected to a keyword node if they use the keyword in their profile description (as described in Section 3). Then we learn the author embeddings in this graph using a two-layer R-GCN with a link prediction objective. In this learning objective, the similarity between two author nodes is maximized if they are connected in the graph and the similarity is minimized if they are not connected in the graph. The link prediction loss function, $L_{link}$ is defined as follows.

$$L_{link} = 1 - sim(a_t, a_p) + sim(a_t, a_n)$$

Here, $sim(a_t, a_p)$ is the similarity between author embeddings $a_t$ and $a_p$, where author, $t$ is connected to author, $p$ in the graph. $sim(a_t, a_n)$ is the similarity between author embeddings $a_t$ and $a_n$, where author, $t$ is **not** connected to author, $n$ in the graph. We measure and define similarity as the dot product between author embeddings. We randomly sample 5 negative examples (two authors are not connected) for each positive example (two authors are connected) in each layer of R-GCN.

We train the R-GCN layers by optimizing the loss $L_{link}$ for 10 epochs. Author nodes are initialized using the RoBERTa-tapt encodings of the author profile descriptions or average embedding of randomly sampled 5 tweets by the author if no profile description is found. The keyword nodes are initialized with their RoBERTa-tapt encodings. All author embeddings are learned in a 50 dimensional space using the 2-layer R-GCN. In this manner, we obtain rich 50 dimensional author embeddings that encode the author retweet relationships and their profile descriptions.

We enhance the multitask baseline described above with these social network-enhanced author embeddings. Note that in the case of the weak supervision setup, the tweets are generated by LLMs and there is actually no author for the tweets. As a result, in the case of the weak supervision setup, we simply average the embeddings of all augmented tweets related to a perspective to get an imaginary author representation for those perspectives.

We concatenate the shared RoBERTa-tapt encodings of the tweets and entities with the corresponding learned author embeddings and use these concatenated representations as inputs for the task-

specific feed-forward neural networks in the multitask model. The rest of the multitask learning objectives and the hyperparameters remain the same as described above. In this manner, the multitask approach with author embeddings becomes comparable to our proposed model as they incorporate the same information (textual and author network) in the learning process.

## D.5 Variations of Our Model

We study our model in the following four variations as shown in Tables 4 and 16.

**(1) Text-Discrete:** In this version, only textual elements such as the tweets and the entities, are considered. The interactions among them are not modeled. For example, the nodes corresponding tweets and entities are not connected using edges. Just initialized node representations are used for classification. This version is comparable to frozen RoBERTa.

**(2) Text as graph:** In this variation, the text is converted to a graph consisting of tweet, entity, and hashtag nodes. The relations among these elements are modeled using edges. Note that no author information is added. Tweet stances are inferred only conditioning on learned tweet node embeddings. No self-inference is done in this variation. In this variation, we train our model for at least 15 epochs and after 15 epochs we stop training if the total loss does not increase for three consecutive epochs.

**(3) Text as graph + Author Network:** In this variation the author network is added to the text-only graph, however, no self-learning is done. In this setting, we train our model for at least 15 epochs and after 15 epochs we stop training if the total loss does not increase for three consecutive epochs.

**(4) Text as graph + Author Network + Self-Learning:** This is the final version of our model where self-learning is added with the combined text and social graph. In this version, we follow the stopping criteria as described in Section D.3.

We run all models (including baselines) 5 times using 5 random seeds. The average weighted F1 scores for the classification tasks for all models with the corresponding standard deviations can be found in Table 16 and the average macro F1 scores are reported in Table 4.

## D.6 Model Implementation Libraries

We use the DGL Graph Library (https://www.dgl.ai/) and PyTorch to implement all of the mod-

els. We use AdamW optimizer for all of the models for optimizing parameters.

### D.7 Infrastructure

We run all of the experiments on a 4 core Intel(R) Core(TM) i5-7400 CPU @ 3.00GHz machine with 64GB RAM and two NVIDIA GeForce GTX 1080 Ti 11GB GDDR5X GPUs. GPT-J-6B was mounted using two GPUs. To run our graph-based model for 300 epochs it takes around 8 hours in this infrastructure.

We submit our model implementation scripts with this manuscript. All of our pretrained models and scripts for implementation will be made publicly available upon acceptance of this paper. A randomly generated subset of the LLM-generated training set and human-annotated test set is submitted with this manuscript for review. We will also publish these datasets by maintaining proper Twitter privacy protocols upon acceptance.

### D.8 Ablation

To determine how many training examples are needed to learn our proposed self-learning approach, we ran an ablation study. We randomly sample 33.33%, 66.67%, and 100% of the training data from the LLM-generated training set (weak supervision) and human-annotated training set (direct supervision) (as summarized in Table 3) and train our proposed model and the multitask baseline (with author embeddings) using the sampled set. We run this sampling and train the models using five random seeds (1000, 2000, 3000, 4000, 5000) and create learning curves using the average over all runs. The learning curves for combined perspective and tweet stance detection are shown in Figure 4. It can be observed that in both weak and direct supervision setups, our model is less sensitive to the number of training data compared to the multitask baseline in both of the tasks. It proves that our proposed model can achieve good performance with a little amount of annotated or artificially generated data.

## E Qualitative Evaluation Details

We infer perspectives using our model and the multitask baseline on the whole #BLM corpus (in Table 1). In the weak supervision setup, we use all the tweets generated by GPT-3 as training data, and in the direct supervision setup, we use all the human-annotated data for training (combining the human-

annotated test and train sets in Table 3). We use a random seed of 1000 for initializing all models and the same hyperparameters as described in Sections D.3 and D.4. The following qualitative evaluations are done using the inferred labels.

### E.1 Correlation of author behavior with stance

To do the analysis on the correlation of authors' following behavior with their stance on #BLM, we first compile a list of tweets accounts of US politicians from the publicly available congressional tweets corpus available on `https://github.com/alexlitel/congresstweets`. Then we collect the follow relationships between these US politicians and the authors in our dataset. Then we measure the Point-biserial correlation coefficient (Tate, 1954) between the percentage of time a user follows a politician from a specific political party (Republican/Democratic) and the identified stance of the author by the models. We find all correlations with $p < 0.0001$. $1,528$ authors are found to follow at least one politician in our corpus.

To analyze the correlation of authors' sharing behavior with their stance, we collect the list of left and right news media domains from `https://mediabiasfactcheck.com/`. Then we measure the Point-biserial correlation coefficient between the percentage of time a user shares an article from a media outlet with a specific bias (left/right) and the identified stance of the author by the models. We find all correlations with $p < 0.0001$. $2,217$ authors are found to share at least one news article from the news sources we gathered. Note that, during preprocessing, all URLs are removed from the tweet texts. Hence, the media outlet domain information is not encoded in the input embeddings in our model.

The numeric values in the correlations bar plot in Figure 5 can be found in Table 17.

### E.2 Entity mapping analysis

First, we manually detect groups of literal entities that are the same. For example. "lives", "their lives" are merged into the literal entity "lives". Then we detect the high PMI perspectives associated with each of these literal entities in each of the camps using the equation described in Appendix A. The high PMI perspectives are reported in Table 6. Some example tweets corresponding to the perspectives in Table 6 and the literal entity groups are shown in Tables 20 and 21, respectively. We find that our

| | Keyword-Based | Multitask (weak sup.) | Multitask (direct sup.) | Our Model (weak sup.) | Our Model (direct sup.) |
|---|---|---|---|---|---|
| Corr(Pro-BlackLM, Follow Democrats) | 0.22 | 0.42 | 0.50 | 0.57 | 0.57 |
| Corr(Pro-BueLM, Follow Republicans) | 0.24 | 0.46 | 0.56 | 0.63 | 0.63 |
| Corr(Pro-BlackLM, Share Left Media) | 0.37 | 0.44 | 0.48 | 0.54 | 0.53 |
| Corr(Pro-BueLM, Share Right Media) | 0.37 | 0.44 | 0.48 | 0.54 | 0.53 |

Table 17: Point-biserial correlation coefficient values between author stance and their following and sharing behaviors on Twitter. All the correlations had a $p$ value $< 0.0001$.

model is able to capture the pattern when the same phrases are used by the different movements to address different entities and also the mapping and sentiments towards some common named-entities (e.g., "Derek Chauvin") better than the multitask baseline as shown in Table 18.

### E.3 Discourse of the movements

Table 19 shows the discourse of the pro-BlackLM and pro-BlueLM camps using perspectives and moral foundations.

To analyze the discourse of the movements we identify the high PMI perspectives associated with each campaign using the same formula as described in Appendix A.

For the identification of moral foundations in the tweets, we train a RoBERTa-based classifier on out-of-domain data. We take the Twitter moral foundation dataset proposed by Johnson and Goldwasser, 2018. In this dataset, 2k tweets are annotated for one or more moral foundations. We encode the tweets using our pre-trained RoBERTa-tapt ([CLS] token of the last hidden layer) and use a fully connected layer to identify the moral foundation labels. This is a multi-label classification task. Hence, we use BCE loss and AdamW optimizer for optimization. We stop training the model if the validation accuracy does not improve for three consecutive epochs. The moral foundation classification F1 score on a held-out validation set was 75.19%. We infer the moral foundation labels in the #BLM corpus using this classifier.

### E.4 Temporal analysis

We calculate the percentage of tweets where a specific perspective is used on a given day. For one type of perspective such as negative actors we calculate the percentage taking tweets only mentioning negative actors in a particular stance. The temporal trends are shown in Figure 8.

## F Literature Review

**Understanding the discourse of the #BLM movement:** The discourse of the #BLM movement is understudied in NLP. One of the early works (Keith et al., 2017) proposed a distantly supervised EM-based approach for Identifying the names of civilians in news articles killed by police. Later Nguyen and Nguyen, 2018 incorporated Deep Learning methods in this task. Both of the works study only named entities and lack the study of how entities are addressed using other words (e.g., "Thugs", "Heroes") or simply using Pronouns. Recently, Ziems and Yang, 2021 introduced the Police Violence Frame Corpus of 82k news articles covering 7k police killings and they studied entity-centric framing where the frame towards the victim of police violence is defined as the age, gender, race, criminality, mental illness, and attacking/fleeing/unarmed status of the victim. All of these works are done on more formal texts such as news articles and do not address how different entities are addressed and what the sentiment and perspectives expressed towards them are.

Recently, a shared task of identifying BLM-centric events from large unstructured data sources has been proposed (Giorgi et al., 2021b) and Giorgi et al., 2022 introduced a large tweet corpus on #BLM paving the way for more studies including ours in this area. In this paper, we unify the identification of co-referenced entities, perspectives toward them in terms of the moral roles assigned to them, and stance prediction on # the BLM movement on highly noisy social media texts.

The discourse of the #BlackLivesMatter movement is mostly studied in social science and computational social science literature. For example, one line of research studied the social dynamics, ties, and in-group commitments that influence the mobilization and formation of the movement. For example, Williamson et al., 2018 found that #BLM protests are more likely to occur in localities where

| Literal Entities | Most assigned perspectives by our model (direct sup.) | | Most assigned perspectives by Multitask model (direct sup.) | |
| | In pro-#BlackLM | In pro-#BlueLM | In pro-#BlackLM | In pro-#BlueLM |
|---|---|---|---|---|
| Black Victims | blacks pos. target | blacks neg. actor | blacks pos. target | blacks neg. actor |
| Derek Chauvin | police neg. actor | N/A | blacks pos. target, police neg. actor | blacks neg. actor |
| Thugs | police neg. actor | antifa neg. actor | racism neg. actor, police neg. actor | antifa neg. actor, BLM neg. actor |
| David Dorn | blacks pos. target | police pos. target | blacks pos. target | police pos. target |
| Lives | blacks pos. target | community pos. target, police pos. target | blacks pos. target | police pos. target |
| Donald Trump | government neg. actor | republicans pos. actor | republicans neg. actor | republicans pos. actor |
| They | blacks pos. target | democrats neg. actor, antifa neg. actor | whites neg. actor, racism neg. actor | democrats neg. actor, antifa neg. actor |

Table 18: Examples of literal entity to abstract entity mapping. Sometimes pro-BlackLM and pro-BlueLM use the same phrase to address different entities and/or perspectives. The lexicon of the literal entities in this table can be found in Table 21.

| DISCOURSE IN PRO-#BLACKLIVESMATTER | | | DISCOURSE IN PRO-#BLUELIVESMATTER | | |
| High PMI Perspectives | MFs in Context | Other Pers. in Context | High PMI Perspectives | MFs in Context | Other Pers. in Context |
|---|---|---|---|---|---|
| police neg. actor | fairness/cheating, authority/subversion | blacks pos. target, blm movement pos. actor | police pos. actor | authority/subversion, loyalty/betrayal | police pos. target, antifa neg. actor |
| blm movement pos. actor | loyalty/betrayal, fairness/cheating | blacks pos. target, racism neg. actor | blm movement neg. actor | authority/subversion, loyalty/betrayal | antifa neg. actor, democrats neg. actor |
| racism/racists neg. actor | fairness/cheating, loyalty/betrayal | blacks pos. target, whites neg. actor | antifa neg. actor | authority/subversion, loyalty/betrayal | blm movement neg. actor, democrats neg. actor |
| blacks pos. target | fairness/cheating, loyalty/betrayal | police neg. actor, blm movement pos. actor | democrats neg. actor | authority/subversion, loyalty/betrayal | community pos. target, antifa neg. actor |
| whites neg. actor | fairness/cheating, auth./subversion | blacks pos. target, racism neg. actor | community pos. target | authority/subversion, loyalty/betrayal | democrats neg. actor, antifa neg. actor |
| blm movement pos. target | loyalty/betrayal, authority/subversion | blacks pos. target, blm pos. actor | police pos. target | loyalty/betrayal, authority/subversion | police pos. actor, antifa neg. actor |
| government neg. actor | authority/subversion, loyalty/betrayal | blacks pos. target, racism neg. actor | republicans pos. actor | loyalty/betrayal, fairness/cheating | democrats neg. actor, antifa neg. actor |
| petition pos. target | fairness/cheating, loyalty/betrayal | blacks pos. target, police neg. actor | | | |

Table 19: Discourse of movements explained with messaging choices and Moral Foundations (MFs). Moral Foundation care/harm was used in all of the cases by both sides. Hence, it is removed from the table. This table is created using the predictions by our model in the direct supervision setup.

more Black people have previously been killed by police, Hong and Peoples, 2021 studied how social ties influence participation in the movement and Peng et al., 2019 discovered that people who join the movement in response to a real-life event are more committed to it compared to the other types. Another line of research studied the power of social media in mobilizing the movement (Mundt et al., 2018; Freelon et al., 2018; Grill, 2021). Some works have studied what properties accelerate mobilization. For example, Casas and Williams, 2019 studied the power of images in mobilizing the movement and Keib et al., 2018 studied what types of tweets are retweeted more related to the movement. De Choudhury et al., 2016 studied temporal analysis of the #BLM movement based on social media, engagement of people from different demographics, and their correlation with textual features.

One major drawback of these studies is that many of them often involve expensive human studies.

There have been several attempts to understand and analyze the narrative in online text in response to the murders of Black individuals in social science. For example, Eichstaedt et al., 2021 and Field et al., 2022 studied the emotions expressed in the online text in response to murders of Black persons. Another line of research studied the framing (Stewart et al., 2017; Ince et al., 2017), rhetoric functions (Wilkins et al., 2019), participation and attention to topics (Twyman et al., 2017), and narrative agency (Yang, 2016) related to the movement. Some other works studied hashtag-based analysis of the #BlackLivesMatter movement (Blevins et al., 2019) and the divergence of the counter-movements (e.g., the #AllLivesMatter movement) (Gallagher et al., 2018).

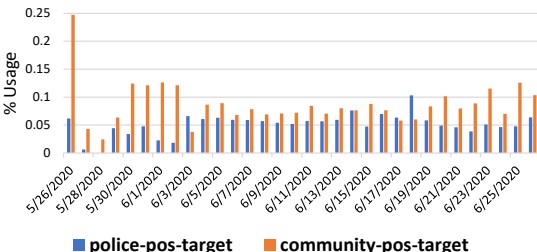

(a) Portrayal of entities as positive targets in pro-BlueLM over time.

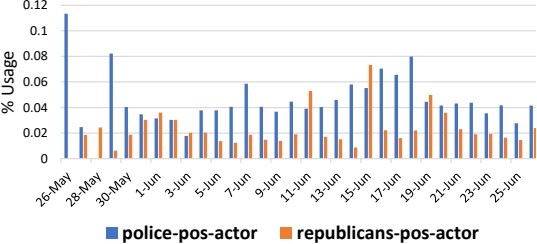

(b) Portrayal of entities as positive actors in pro-BlueLM over time.

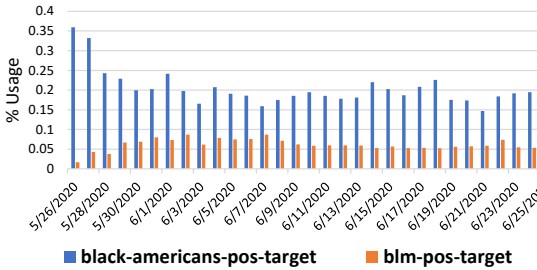

(c) Portrayal of entities as positive targets in pro-BlackLM over time.

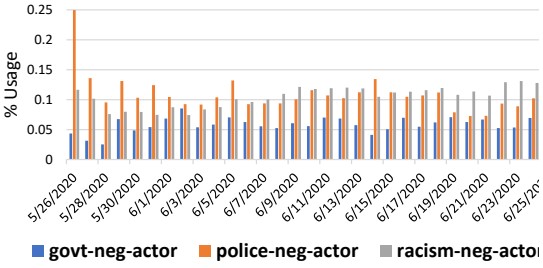

(d) Portrayal of entities as negative actors in pro-BlackLM over time.

Figure 8: Temporal trends in all camps for different types of entity roles identified using our model in the direct supervision setup. Note that, mostly BLM is portrayed as positive actors in pro-BlackLM and the portrayal of other entities as positive actors is significantly lower in this camp. Hence, the trend for positive actors in pro-BlackLM is not shown in this analysis.

| Literal Entities | Example tweets in pro-#BlackLivesMatter | Example tweets in pro-#BlueLivesMatter |
|---|---|---|
| Black American Victims | Arrest the cops that killed **BreonnaTaylor**! #BlackLives-Matter | Thanks for the body cam. It is obvious to anyone, but #Liberals that **RayshardBrooks** was violently resisting arrest. #Atlanta #AllLivesMatter |
| Derek Chauvin | **Derek Chauvin**, the murderer policeman who killed George Floyd, would not have been arrested had it not been for the uprising in Minnesota. | N/A |
| Thugs | The behaviour of the American police is absolutely revolting. Racist yobs with military grade weapons empowered by a fascist president. Why should we automatically be expected to "respect" these **thugs**? Respect should be earned. #BlackLivesMatter | BlackLivesMatter is the black version of the white trash Antifa **thugs**. They could care less about black lives, they're **thugs** and trash. #RememberDavidDorn #AllLivesMatter |
| David Dorn | This isn't political, all #BlackLivesMatter. **David Dorn**, #AhmaudArbery, #GeorgeFloyd #BreonnaTaylor. Stop turning this into something else for a reaction. | Where are you all when **#DavidDorn** was shot in the head by #BlackLivesMatter rioters and #AntifaTerrorist ? Disgusting silence. |
| Lives | The countless black **lives** that have been lost in the U.S. due to acts of #policebrutality have been tolerated & condoned by the government enough to be considered Crimes against humanity imo. I think America should be held accountable for this on an Int. level. #BlackLivesMatter | The Police should leave. It's not worth their **lives** for this BS anymore. #BlueLivesMatter |
| Donald Trump | **@realDonaldTrump** should get accustomed to being surrounded by fencing & prison guards. #protests #BLM #BunkerBoyTrump #Bunkerbaby #TrumpIsAnIdiot #TrumpIsACoward #DCProtests #ArrestTrump #BlackLivesMatter | **@realDonaldTrump** has asked for Unity from Day 1 unlike the DemoKKKrats And @TheDemocrats and their leaders are the one calling for violence, not **Trump** |
| They | HOW ARE THESE CASES NOT MAKING HEADLINES?! **They** are being murdered and nothing is being done about it. #BlackLivesMatter | And this is what **they** want to reform!??! Morons cannot win! |

Table 20: Examples of tweets where the pro-BlackLM and pro-BlueLM use the same phrases to address different entities and/or perspectives. The literal entities in the example tweets are bolded and underlined. The lexicon of the literal entities in this table can be found in Table 21. "N/A" means the entity is not mentioned frequently in a particular campaign.

| Literal Entities | Lexicon | Description |
|---|---|---|
| Black American Victims | # georgefloyd, george floyd, georgefloyd, # breonnataylor, breonna taylor, breonna, # georgeflyod, george, # george-floyd #, # rayshardbrooks, rayshard brooks, rayshardbrooks, # ahmaudarbery, ahmaud arbery, robert fuller, samuel du-bose, sandra bland, walter scott | Names of the Black American persons who were killed. |
| Derek Chauvin | derek chauvin, chauvin | The police officer who killed George Floyd. |
| Thugs | thugs, these thugs, the thugs | A negative term to address a person or a group of persons. |
| David Dorn | # daviddorn, david dorn | Black Police officer who was killed during the George Floyd protests. |
| Lives | their lives, lives, the lives, our lives, life, my life, his life, her life, their life, the life, a life | Self-explanatory. |
| Donald Trump | @ realdonaldtrump, @realdonaldtrump, realdonaldtrump, trump supporters, trump, # trump, president trump, donald trump, trump2020 | 45th U.S. President who is from the Republican Party. |
| They | they | The Pronoun they. |

Table 21: Description of the literal entities studied in the qualitative evaluations.

The existing studies in computational social science on narrative understanding mostly rely on hashtag or lexicon-based analysis of the movements, hence, they fail to capture the nuances in the messaging choices and often cannot differentiate between movements and counter-movements when representative hashtag from one movement is "hijacked" (Gallagher et al., 2018) by the supporters of the counter-movement.

In contrast to these existing studies in CSS and NLP on #BLM, in this paper, we propose a holistic technical framework for characterizing such social movements on online media by explicitly modeling the perspectives of different camps.

**Perspective Analysis and Stance Detection:** Revealing perspectives in complex and deceptive discussions is an important part of discourse analysis and it has been studied in different settings and variations in recent studies. For example, Thomas et al., 2006 and Burfoot et al., 2011 attempted to identify stances in congressional floor-debate transcripts (against or in support of proposed legislation). Another line of research studied stances in online debate forums where the stance (pro vs. con) of a speaker on a specific issue is predicted (Somasundaran and Wiebe, 2010; Walker et al., 2012; Hasan and Ng, 2013, 2014; Sridhar et al., 2015; Sun et al., 2018; Xu et al., 2018; Li et al., 2018). Mohammad et al., 2016 introduced a shared task of predicting stances in microblogs such as tweets where the task is to identify the stance in a tweet with respect to a given target (e.g. entity, issue, and so on). Consequently, more recent works have studied political stances in politically controversial issues (Johnson and Goldwasser, 2016b,a; Ebrahimi

et al., 2016; Augenstein et al., 2016). Recently, another shared task has been proposed (Pomerleau and Rao, 2017) where the task of fake-news detection is studied from a stance detection perspective. In this task, a headline and a body text are given - either from the same news article or from two different articles. The task is to determine if the stance of the body text relative to the claim made in the headline is in agreement, disagreement, discussion, or irrelevant.

Identification of stance is studied using various approaches (Cignarella et al., 2020). For example, Burfoot et al., 2011 and Sridhar et al. (2015) studied stance detection using a collective classification approach. Sridhar et al., 2015 and Hasan and Ng, 2014 studied not only stances but also the reasoning behind stances and the disagreements among them. In a related work in this line, Somasundaran and Wiebe, 2010 used lexicon-based features for detecting "arguing" opinions, and supervised systems using sentiment and arguing opinions were developed for stance classification. Sun et al., 2018 proposed a neural model to learn mutual attention between the document and other linguistic factors. Thomas et al., 2006 leveraged inter-document relationships and Walker et al., 2012 leveraged the dialogic structure of the debates in terms of agreement relations between speakers for stance detection. Another line of research is built on relational learning-based methods for stance detection where various types of contextualizing and relational properties are explicitly modeled and incorporated in stance detection. For example, Johnson and Goldwasser, 2016b proposed a relational learning framework incorporating framing and temporal activity patterns, and

Ebrahimi et al., 2016 proposed a relational model incorporating the friendship networks of the authors. The main drawback of such relational learning approaches is that all possible solutions to the problem are explored and the one with the best gain is returned, as a result, the inference tree becomes really large and computationally expensive for a large amount of data.

For the analysis of perspectives in polarized topics, different socio-linguistic theories have been used in the literature. For example, the Moral Foundation Theory (MFT) (Haidt and Joseph, 2004; Haidt and Graham, 2007) and framing analysis (Entman, 1993; Chong and Druckman, 2007; Boydstun et al., 2014). Framing is referred to the approach of communication by focusing on certain aspects of a story in order to bias the readers toward certain stances. Previous studies used framing to understand political perspectives and communication strategies by biased news sources and social media users (Tsur et al., 2015; Baumer et al., 2015; Card et al., 2015; Field et al., 2018; Demszky et al., 2019; Fan et al., 2019; Roy and Goldwasser, 2020). The Moral Foundation Theory is also widely used for analyzing perspectives (Dehghani et al., 2014; Fulgoni et al., 2016; Brady et al., 2017; Hoover et al., 2020; Roy and Goldwasser, 2021). However, Moral Foundations are widely used to understand sentence level perspectives. Roy et al. (2021) introduced Morality Frames which is a knowledge representation framework for capturing entity-centric moral sentiments. Because of the expressivity of entity-centric moral foundations and their strong correlation with stances, in this paper, we use morality frames for modeling perspectives in the #BlackLivesMatter and #BlueLivesMatter movements.

**Data Augmentation Approaches in NLP:** Data augmentation refers to the technique where additional and ideally diverse data are generated without explicitly collecting new data. Data augmentation techniques can be useful in a low-resource setting where obtaining dataset for training models are difficult. Annotated data for nuanced perspectives are difficult to get as annotating them pertains to specialized knowledge. Hence, automatic data augmentation is desirable.

With the recent advances in deep learning-based models, different data augmentation techniques have been proposed. For example, Kobayashi, 2018 proposed RNN-based, Kumar et al., 2019

trained sequence to sequence models, and Yang et al., 2020 and Ng et al., 2020 proposed pretrained transformer-based approaches for data augmentation. In most of these approaches, existing text data is modified to augment new data. In contrast to these studies, Anaby-Tavor et al., 2020 and Quteineh et al., 2020 directly estimated the text generation process using GPT-2 (Radford et al., 2019) rather than modifying an existing example to generate new data.

With the recent advances of transformer-based pretrained auto-regressive Large Language Models (LLMs) such as GPT-3 (Brown et al., 2020), a new direction for data augmentation has opened up where controlled text generation (Iyyer et al., 2018; Kumar et al., 2020; Liu et al., 2020) is feasible by prompting these pretrained LLMs. These LLMs are trained on a huge corpus of web crawls, hence, have the capability to generate human-like text. One recent work by Liu et al., 2022 combined the generative power of these LLMs and the evaluation power of humans and proposed a human-LLM interaction loop to generate datasets for Natural Language Inference (NLI) tasks. Inspired by these recent advances in data augmentation using LLMs, in this paper, we propose to augment few-shot training data using LLMs by prompting them in a structured way. This augmented dataset is later used to bootstrap our proposed model.