# OpenReview forum: ""A Tale of Two Movements": Identifying and Comparing Perspectives in \#BlackLivesMatter and \#BlueLivesMatter Movements-related Tweets using Weakly Supervised Graph-based Structured Prediction"
_EMNLP/2023/Conference — EMNLP 2023 Findings_

### Official Review · Reviewer_qRYV · 2023-08-01

**Soundness:** 4

**Ethical Concerns:**

Yes

**Excitement:**

2: Mediocre: This paper makes marginal contributions (vs non-contemporaneous work), so I would rather not see it in the conference.

**Justification For Ethical Concerns:**

Black-box proprietary models that one can only access through an API should not be accepted as part of valid scientific methodology and such behavior should not be normalized.

**Paper Topic And Main Contributions:**

In this work the authors propose a graph-based learning approach to jointly learning socio-semantic representations of the opposing "Black Lives Matter" and "Blue Lives Matter" movements as expressed in a corpus of tweets. The authors define perspectives by adapting Morality Frames theory and capturing two dichotomies about identified abstract entities: actor / target (having agency or being acted upon) and sentiment (positive / negative). A multi-relational graph is considered, combining relations between author and tweet, author and author, author and keyword, hashtag and tweet and entity and tweet.

A self-learning loop is employed, where the training set is iteratively augmented with tweets and entities that are deemed to be "consistent". Shortly, a tweet is consistent if its predicted stane is aligned with the predicted perspectives of the entities mentioned in the tweet and an author is deemed consistent if all of their consistent tweets have the same alignment. Furthermore, the authors propose a "weakly supervised" approach where synthetic training data is generated with GPT-3, as a strategy to mitigate the effort inherent in annotating large text corpora. In all cases, models are tested with human-labelled data.

F1 scores show that the framework proposed outperforms multitask RoBERTa and ablation tests show that the representation of text as graphs of entities, the use of the author network and the self-learning loop all contribute to higher scores. Common perspectives on various entities from both Black Lives Matter and Blue Lives Matter are presented, and they agree with one's intuitive understanding of theses movements.

**Reasons To Accept:**

This work focus on a very important contemporary topic: the understanding of opposing political movements within the context of a highly polarized society and mass communication through social media. The characterization of perspectives is grounded on Morality Frames theory from the field of linguistics and translates it successfully to a deep learning based NLP framework.

The deep learning model is technically sound, the choices are well justified and the experiments are performed in a rigorous fashion.

**Reasons To Reject:**

Unfortunately, I have one major objection to one aspect of the methodology, that I believe makes this work unsuitable for publication in a scientific conference: the use of a proprietary, black-box LLM or the generation of synthetic training cases. It is completely unacceptable to rely on proprietary black-boxes for scientific research, and if we allow for this we risk doing irreparable damage to the field. We can't really know what the model behind the API is, nor can we know how it was trained. It seems fairly obvious that GPT-3 training corpus included tweets, but we cannot know which ones, selected according to what criterion. This "weakly supervised" approach is in fact extracting knowledge from a black box. I have no doubt that this worked well in this case, as the authors report, but this is useless in a scientific setting.

I imagine that the aspect of the article that relies on GPT-3 could be extracted, but even then I believe that big issues remain. The underlying assumptions of the self-learning loop are fairly strong: that a tweet must not express ambiguous / ambivalent stances about the issue and that an author never changes their mind. Such assumptions clearly work well in producing consistency, and it looks that it works well for this specific case, but it doesn't really produce any insight in the end that couldn't be quickly obtained by human annotators inspecting a sample of tweets on both sides. Could it be that introducing such strong consistency requirements throws the baby with the bath water? Won't it guarantee that nothing interesting is actually detected? After all, the point of automating text understanding is to produce insights that could not be obtained by traditional methods with a small amount of effort.

What remains is the insight that text-as-graph and social features improve the quality of such models. Ok, but this is already well known in the field on a variety of use-cases, for example:

Fake user detection on Twitter:
Balaanand, Muthu, et al. "An enhanced graph-based semi-supervised learning algorithm to detect fake users on Twitter." The Journal of Supercomputing 75 (2019): 6085-6105.

Sentiment analysis on Twitter:
Lovera, Fernando Andres, Yudith Coromoto Cardinale, and Masun Nabhan Homsi. "Sentiment Analysis in Twitter Based on Knowledge Graph and Deep Learning Classification." Electronics 10.22 (2021): 2739.

Detecting toxicity on Twitter:
Jiang, Jian, Rui Wang, and Guo-Wei Wei. "GGL-Tox: geometric graph learning for toxicity prediction." Journal of chemical information and modeling 61.4 (2021): 1691-1700.

**Reproducibility:**

1: Could not reproduce the results here no matter how hard they tried.

**Reviewer Confidence:**

4: Quite sure. I tried to check the important points carefully. It's unlikely, though conceivable, that I missed something that should affect my ratings.

---

> ### Author Rebuttal · Authors · 2023-08-28
>
> Thank you for your review!
>
> $\textbf{GPT-3 usage:}$ Usage of GPT-3 is neither the central theme of our paper, nor our model solely depends on GPT-3. The main focus of this work is to disentangle the discourses on the #Blacklivesmatter and #Bluelivesmatter movements by explicitly modeling the perspectives in them. Our proposed model requires only a few training data samples to identify perspectives in a large social media corpus. This little amount of training data can be – artificially crafted or real data annotated by humans. To support this claim we performed all our experiments separately using GPT-3 generated and human-annotated training sets and tested only on human-annotated test data. Even if we drop the results that we obtained using the GPT-3 generated training data, the main claims and findings of our paper do not change at all. Also, we do not believe, the usage of GPT-3 for such a non-central and small aspect of our model makes our whole modeling approach non-reproducible.
>
> We prompted GPT-3 to produce some coherent text given a stance, entity, hashtag, and moral sentiment towards the entity. Hence, we are not extracting any additional information from the LLM. Moreover, one author of this paper manually went through the generated few-shot examples to observe if any additional information was added to the tweets. Please refer to section 4, Appendix B for the method and Table 11 for generated examples.
>
> The capability of GPT-3 in various tasks is being explored widely in recent scientific works and they are being accepted in highly reputed scientific publication venues (e.g., ACL, NeurIPS, etc.). Hence, we believe the way we used GPT-3 in this paper is not at all non-scientific. We are listing a couple of recent papers that used GPT-3 and were published in ACL and NeurIPS below for reference.
>
> - “When to Make Exceptions: Exploring Language Models as Accounts of Human Moral Judgment.” Jin et al. NeurIPS 2022
>
> - “Does GPT-3 Grasp Metaphors? Identifying Metaphor Mappings with Generative Language Models”. Wachowiak and Gromann. ACL 2023.
>
>
>
> >The underlying assumptions of the self-learning loop are fairly strong: that a tweet must not express ambiguous / ambivalent stances about the issue and that an author never changes their mind.
>
> We are dealing with highly polarized movements and the source dataset we used contains only the tweets that mentioned support-indicating keywords (#Blacklivesmatter, #Bluelivesmatter), not neutral cases. To infer stances and perspectives in such polarized movements, we deployed the self learning loop based on consistency in authors' stance and perspective. However, the self-learning loop does not prevent us from finding ambiguous and interesting cases. For example, when the supporters of a specific movement use the keyword of the opposite movement to "hijack" the discussion (supported by previous work by Gallagher et al., 2018) (L391-L397). This is studied in depth in our paper and our finding is that in such ambiguous cases also, our model excels (Table 4, L460-L494). Because, even when the keywords/hashtags are hijacked, the author’s social media interaction and perspective toward the main actors, help disambiguate such deceptive tweets, indicating the effectiveness of the assumptions in the self-learning loop.
>
> We studied the discourse of the #BlackLivesMatter and #BlueLivesMatter movements in the timeframe of 1 month after George Floyd's killing. It is highly unlikely that an author’s stance will change in such a short period. We acknowledged such assumptions in the Limitations section and will make it clearer.
>
> > it doesn't really produce any insight in the end that couldn't be quickly obtained by human annotators inspecting a sample of tweets on both sides
>
> As stated above and thoroughly studied in the paper, our main focus is to disentangle opposing movements by explicitly modeling and inferring perspectives in a large social media corpus (400k tweets) by using only a few training examples (~500 tweets). We do not think, identifying perspectives in such a large corpus is feasible with human annotation as it involves difficulties in entity disambiguation and stance inference that requires more context than just text and specialized human knowledge. This claim is explored in depth by the experiments presented in the paper.
>
> >What remains is the insight that text-as-graph and social features improve the quality of such models. Ok, but this is already well known in the field on a variety of use-cases
>
> To the best of our knowledge, none of the prior works (including the ones you mentioned) studied text-as-graph in the same spirit as we did, and none combined text-as-graph and the social media interaction graph. However, we took inspiration from prior influential works that used social network information in text understanding (however, mostly studied text as a single unit) in NLP and are adequately cited in our paper (L616-L624).
>
> >Fake user detection on Twitter: Balaanand, Muthu, et al. "An enhanced graph-based semi-supervised learning algorithm to detect fake users on Twitter." The Journal of Supercomputing 75 (2019): 6085-6105.
>
> This paper does not really study textual content in depth, rather interactions of users with tweets such as fraction of retweets, average time between tweets, and some shallow textual features like tweet length, etc. are used as features to classify fake users.
>
>
> > Sentiment analysis on Twitter: Lovera, Fernando Andres, Yudith Coromoto Cardinale, and Masun Nabhan Homsi. "Sentiment Analysis in Twitter Based on Knowledge Graph and Deep Learning Classification." Electronics 10.22 (2021): 2739.
>
> The authors in this paper extracted the dependency graph (PoS tags) in text by using a dependency parser (and addressed it as knowledge graph) and used that for sentiment analysis which is absolutely not related to what we mean by text-as-graph in our paper. Our focus is on a rich semantic graph representation for text consisting of entities, moral sentiment towards entities, stance indicator hashtags, and connecting that with the social network interaction of the authors for getting a rich contextualized representation of text (Section 3.1). The above paper did not include any social interaction, and the authors did not even compare their model with other state-of-the-art NLP models at that time. Most concerningly, it was published in a journal that was listed as predatory, and accused of self-citation, and treachery many times in the last 10 years (currently it is listed as a predatory journal in at least one ranking).
>
> >Detecting toxicity on Twitter: Jiang, Jian, Rui Wang, and Guo-Wei Wei. "GGL-Tox: geometric graph learning for toxicity prediction." Journal of chemical information and modeling 61.4 (2021): 1691-1700.
>
> We want to respectfully indicate that this paper you suggested is not a study on toxicity on Twitter, rather it is an article related to drug design and discovery, and the authors apparently studied toxicity in environmental chemicals and drugs.

---

### Official Review · Reviewer_GbZ4 · 2023-08-05

**Soundness:** 5

**Excitement:**

4: Strong: This paper deepens the understanding of some phenomenon or lowers the barriers to an existing research direction.

**Paper Topic And Main Contributions:**

This paper develops a methodology for identifying perspectives in tweets related to the Black Lives Matter movement. They focus on two stances: pro-BlackLivesMatter (pro-BlackLM) and pro-BlueLivesMatter (pro-BlueLM). They define perspectives in terms of sentiment (positive/negative) toward entities (eg, Black Americans, Police) and their roles (actor/target).

To identify perspectives, they develop a graph-based methodology that unifies a number of prediction tasks. First, they construct a graph representing the tweets (with tweet-entity and tweet-keyword relations) and authors (with author-tweet and author-author relations). They use relational GCN (for multiple relation types) to learn representations of the nodes. Then they define a series of prediction tasks--tweet stance prediction, entity sentiment and role prediction, and entity stance prediction--that are all predicted from the node embeddings, thus unifying the tasks under a common framework. Finally, they incorporate self-learning, where the model begins with a small amount of annotated data, then learns to add training data based on consistency (eg, a tweet stance that is pro-BLM is consistent with the perspective police-neg-actor).

To train the models, they try direct supervision and weak supervision. For direct supervision, they sample 200 users and manually annotate their tweets. For weak supervision, they generate tweets using GPT-3. They try training with real train data and generated data, and evaluate on real test data in both cases. They find that their model outperforms baselines and that each aspect of their method (representing tweets as graph, adding the author relations to the graph, adding self-learning) improves the model's performance. Direct supervision outperforms weak supervision, but sometimes not by much. Finally, they conduct some qualitative analyses and find expected correlations, eg, between stance and political affiliation, and they find their correlations are stronger for their model than for baselines.

**Questions For The Authors:**

A. How generalizable is your method - ie, how much work would a new researcher need to do (eg, in defining entities) to apply this method to another social phenomena with multiple perspectives?

**Reasons To Accept:**

- The authors introduce an innovative methodology that jointly performs a number of tasks related to perspective identification. While the graph neural network is standard (R-GCN), it is innovative to define several prediction tasks from the same node embeddings and to capture their dependencies by training them jointly and enforcing consistency.
- Their model achieves strong performance and outperforms baselines, and many aspects of their method (representing tweets as graph, adding the author relations to the graph, adding self-learning) improve the model performance.
- The task is challenging and interesting - identifying differing stances on the same topics - and shows up frequently in analyses of social data and social phenomena.

**Reasons To Reject:**

- Weak supervision could be better evaluated - eg, how realistic are the evaluated tweets? The prompt requires "all of the structured elements for perspectives to be present in the generated tweets", which doesn't see the most realistic. The generation of authors is also not realistic ("[author] embeddings are initialized by averaging the corresponding artificial tweets").
- The authors also claim that weak supervision achieves "comparable" performance - this feels like a bit of an overstatement. For author stance, performance drops 5 points from direct to weak (same gap as from their model to baseline); also substantial drops for ambiguous (30 points) and entity mapping (8 points).

**Reproducibility:**

5: Could easily reproduce the results.

**Reviewer Confidence:**

4: Quite sure. I tried to check the important points carefully. It's unlikely, though conceivable, that I missed something that should affect my ratings.

---

> ### Author Rebuttal · Authors · 2023-08-28
>
> Thank you for your review!
>
> > Weak supervision could be better evaluated - eg, how realistic are the evaluated tweets? The prompt requires "all of the structured elements for perspectives to be present in the generated tweets", which doesn't see the most realistic. The generation of authors is also not realistic ("[author] embeddings are initialized by averaging the corresponding artificial tweets").
>
> We found the generated tweets very realistic. Please refer to Table 11 for generated tweet examples. Also, a sample of the tweets is submitted with this manuscript. The structured elements required in the prompts are well defined, hence, we believe it is realistic. The dummy author nodes were generated only in the setting when the tweets were augmented using LLM; when a real author exists, this step is not required in our model.
>
> > The authors also claim that weak supervision achieves "comparable" performance - this feels like a bit of an overstatement. For author stance, performance drops 5 points from direct to weak (same gap as from their model to baseline); also substantial drops for ambiguous (30 points) and entity mapping (8 points).
>
> In the other tasks, weak supervision achieves pretty comparable performance to direct supervision and it outperforms most direct supervision baselines. In the immediate next sentences, we discussed the reasons for the failure of weak supervision in the addressed tasks (L489-L494). Regardless, we will rephrase this statement.
>
> > A. How generalizable is your method - ie, how much work would a new researcher need to do (eg, in defining entities) to apply this method to another social phenomena with multiple perspectives?
>
> We believe our method is generalizable to any social phenomenon. The new researchers just have to put a one-time effort into identifying the main actors (defined as abstract entities in our paper) related to the phenomena if the actors are not already given. As discussed in our limitations section (L666-L672), the main actors related to a movement can be extracted from an external database, or by doing a one-time human-in-the-loop data analysis as we did in this paper. The other properties of perspectives such as pro/anti-stances, moral sentiments, etc. are already well defined in the literature.

---

### Official Review · Reviewer_Yvsx · 2023-08-05

**Soundness:** 4

**Excitement:**

2: Mediocre: This paper makes marginal contributions (vs non-contemporaneous work), so I would rather not see it in the conference.

**Paper Topic And Main Contributions:**

This work proposes a weakly supervised graph based approach to determine perspectives during the two movements.  LLMs are used to generate training data. The authors suggest a structured representation  for analyzing online content.

The authors note that disambiguating stances towards a movement is important and they used graph-based representations for it. To account for the annotation costs, they use LLMs to generate synthetic tweet like data. The authors considered tweets posted in the month immediately following a triggering event. They use morality frames to determine the sentiment towards an entity

The authors argue that the stance of an author, the entity disambiguation and entity role are interdependent. The authors use a 2-layer R-GCN for the graph representation. The authors then propose self-learning in order to allow their models to learn constantly.

The training data is generated by prompting LLMs. However, there is no mention of exactly how this data of ~500 tweets was split into pro blm/bluelm? The authors also then go on to perform the experiments with their modeling setups. They find that using their graph techniques and a self-learning framework yields superioir results to a baseline model. However, all evaluations are done using seq2seq models even though LLM inference has been used and could potentially act as a very good baseline for classifying tweets.

The authors also note that the dataset is heavily skewed. This makes one question some of the analysis as making extrapolations based on skewed datasets might not hold.

**Questions For The Authors:**

The self-learning explanation can be clearer - for instance, it is not clear if the number of steps and epochs refer to the same k value. Could this be made clearer in the paper?

The tweet consistency again exposes the flaws that people will always be either completely pro or against movements, there is no indication of how neutral events were handled. Was there a consideration to look at the data and consider what to do with tweets/accounts which potentially had neither positive nor negative sentiment towards the event?

**Reasons To Accept:**

1. This work presents a study on a less-studied dataset.
2. The authors make use of Graph-neural networks to create a graph using texts to analyze social movements on social media websites which is unique.
3. The paper is well-written and presents results clearly.

**Reasons To Reject:**

1. The paper suffers from putting people into exactly two buckets - either pro or against and makes a lot of extrapolations and conclusions based on a black or white interpretation of people's views. This is a really big weakness as the authors could have addressed this by either filtering neutral tweets or adding additional guardrails and verification to throw out non extreme accounts.
2. A number of conclusions are drawn from the analysis on this dataset - however, the authors themselves mention that the dataset is skewed and leans heavily towards pro BlackLM. Some of the assumptions such as all pro blackLM follow democrats or otherwise does not hold much weight when the dataset is skewed.
3. A large portion of the paper is heavily targeted at an audience which is familiar with the American political and news ecosystem. As a result, a large number of world readers might not be completely familiar with the topic.

**Reproducibility:**

3: Could reproduce the results with some difficulty. The settings of parameters are underspecified or subjectively determined; the training/evaluation data are not widely available.

**Reviewer Confidence:**

4: Quite sure. I tried to check the important points carefully. It's unlikely, though conceivable, that I missed something that should affect my ratings.

---

> ### Author Rebuttal · Authors · 2023-08-28
>
> Thank you for your review!
>
> > The paper suffers from putting people into exactly two buckets - either pro or against and makes a lot of extrapolations and conclusions based on a black or white interpretation of people's views.
>
> > The tweet consistency again exposes the flaws that people will always be either completely pro or against movements, there is no indication of how neutral events were handled. Was there a consideration to look at the data and consider what to do with tweets/accounts which potentially had neither positive nor negative sentiment towards the event?
>
> In this paper, we analyzed two extremely polarized movements namely #Blacklivesmatter and #Bluelivesmatter. It seems unlikely, counter-intuitive, and probably outlier cases when people use the phrases "Black lives matter" or "Blue lives matter" and remain neutral. In Computational Social Science studies also, mention of these two phrases is widely considered as endorsement/support or interaction/participation to the corresponding movements (please refer to the papers we cited in the literature review (L2053-L2105)), if not "hijacked" (Gallagher et al., 2018). This is also well-supported by our experiments where the classification of tweet stances based on these keyword mentions yields a pretty high F1 score of 87.77% (Table 4, L460-L467) in human-annotated test data. We want to reemphasize that the source dataset (Giorgi et al., 2022) we used for this study collected $\textbf{only}$ the tweets that mention these stance indicator phrases, hence, they already have guardrails for neutral cases if there are any. Moreover, we took into account the users who tweeted at least 5 times after the killing of George Floyd (L118-L120). It ensures that only high-vocal supporters of the movements are analyzed in this paper.
>
> In real life, there might be neutral users who do not participate in such movements, however, in this paper, our goal is to study the movements (not the neutral cases) and the perspectives that appear inside the movements.
>
> We did not get any mention of such neutral cases from the human annotators during the in-house human annotation process as well. Also, all of our qualitative evaluations are reported at an aggregate level and quantitative evaluations are done using human annotated test set. As a result, even if the slightest amount of neutral-stance outliers are present in the data after all the guardrails, they should not affect the trends we observed during our evaluation. Hence, we believe, we did not extrapolate anything, rather honestly reported just what we observed.
>
> There may be other movements where pro, anti, or neutral stances are almost equally common, and extending our model to those cases should be straightforward. In that case, we can just account for one additional stance that is "neutral" during modeling. Based on your review, we can make these statements clearer in our writing.
>
> >A number of conclusions are drawn from the analysis on this dataset - however, the authors themselves mention that the dataset is skewed and leans heavily towards pro BlackLM ...
>
> We believe the distribution of the #Blacklivesmatter and #Bluelivesmatter tweets in the source dataset (Giorgi et al., 2022) represents real life. It has been well supported by many previous studies and survey reports that #Blacklivesmatter is a much bigger movement and its activists are way more active than #Bluelivesmatter on Twitter. For example, the sharp skewness towards the #Blacklivesmatter tweets is reported by two reputed research institutes as follows.
>
> [Article from Pew research](https://www.pewresearch.org/internet/2018/07/11/an-analysis-of-blacklivesmatter-and-other-twitter-hashtags-related-to-political-or-social-issues/)
> - "From July 18, 2016 through May 1, 2018, the #BlackLivesMatter hashtag has been used an average of 15,856 times daily. The #BlueLivesMatter hashtag has been used an average of 3,998 times daily”.
>
> [Article from Brookings.edu](https://www.brookings.edu/articles/how-george-floyd-changed-the-online-conversation-around-black-lives-matter/)
> - “the #BlackLivesMatter hashtag has emerged as an enduring feature of online discourse. As of April 30, 2021, it has been used in more than 25 million original Twitter posts, which collectively have garnered approximately 444 billion likes, retweets, comments, or quotes—roughly 17,000 engagements per post.”
> - “Between 2013 and 2021, #BlueLivesMatter has registered 1.6 million original posts and 1.7 billion engagements”;
>
> >Some of the assumptions such as all pro blackLM follow democrats or otherwise does not hold much weight when the dataset is skewed.
>
> The point-biserial correlation (L1937-L1970) we used to obtain the correlation between an author’s stance label and their following and sharing behavior, $\textbf{does not}$ depend on the distribution of the dichotomous variable which is the stance of the authors in our case (either pro-BlackLM or pro-BlueLM), which is under discuss for its skewness. The continuous variables (percentage of following of Democrats/Republicans and percentage of sharing from Left/Right media) are normally distributed in our case inside each dichotomous class verified by a Shapiro-Wilk test (although strict normal distribution is not required for point-biserial). [Reference linked.](https://statistics.laerd.com/spss-tutorials/point-biserial-correlation-using-spss-statistics.php)
>
> We found all the correlations with high statistical significance (L1937-L1970), hence, we believe they are sound.
>
> Some other qualitative evaluations are done using pointwise mutual information scores, which also take care of class distributions in the formula (Appendix E.2, E.3).
>
> Also, in our models, we used weighted loss functions and we did a fine-grained evaluation with a human-annotated test set and we presented macro (Table 4), weighted (Table 16), and per class (Table 5) F1 scores for perspectives identification to account for the skewness in any of the classes.
>
> Hence, we do not believe the skewness in the data makes any of our analyses of less weight.
>
>
> >there is no mention of exactly how this data of ~500 tweets was split into pro blm/bluelm?
>
> The LLMs were prompted to generate tweets having either pro-BlackLM or pro-BlueLM stances along with the perspectives (Figures 5 and 6) and this method is clearly described in the paper (Section 4: L346-L361, Appendix B). Hence, the generated tweets are readily split into pro blm/bluelm labels after prompting.
>
> >However, all evaluations are done using seq2seq models even though LLM inference has been used and could potentially act as a very good baseline for classifying tweets.
>
> We are not sure what is meant by "all evaluations are done using seq2seq models". We used RoBERTa-based (an encoder-only model) end-to-end and multitask models as baselines and human-annotated test data for evaluating the classification tasks. We used LLMs only for a few training data generations. Note that generative LLMs by design generate sequences, not class probabilities. Adapting them for classification tasks requires additional effort and extensive prompt engineering, particularly in problems like the one studied in this paper where more context beyond the text is needed and multiple classification decisions are needed to be made jointly. To keep our contribution focused and within the space limit, we leave it as a future work.
>
> >it is not clear if the number of steps and epochs refer to the same k value. Could this be made clearer in the paper?
>
>
> Yes, step and epoch refer to the same thing. We will make it clear.
>
> > A large portion of the paper is heavily targeted at an audience which is familiar with the American political and news ecosystem. As a result, a large number of world readers might not be completely familiar with the topic.
>
> The #BLM protests after the killing of George Floyd in the US gained much attention and spread internationally. The source dataset contains geo-tagged tweets related to #Blacklivesmatter and counter-movements from all over the world and in different languages (please refer to Georgi et al., 2022). We only focused on tweets written in English (not limited to the U.S. only) to keep our study focused (L118). However, we think our model can be applicable to any national or international movement. Hence, we do not think the movements studied and the models proposed in this paper are of limited interest.

---

### Meta-Review · Area_Chair_GaWg · 2023-09-15

**Recommendation:** 4

**Metareview:**

Thank you to the authors for their work and the reviewers for their reviews. This work focuses on automatically generating perspectives about abstract entities in polarized social media text, focusing on #BlackLivesMatter and #BlueLivesMatter tweets.

Pros:
- All reviewers rate the paper high for soundness
- As noted by GbZ4 "The task is challenging and interesting - identifying differing stances on the same topics - and shows up frequently in analyses of social data and social phenomena." This methodology has the potential to be broadly useful for social text analysis
- GbZ4 and Yvsx agree that the graph-based methodology with several prediction tasks is innovative

Cons:
- Several reviewers raise concerns about how the authors do not allow for neutral or ambiguous stances. As the #BlackLivesMatter movement has been highly polarized and the authors remove any users who tweet about the events <5 times, I don't see this as a fundamental flaw in this particular case study. However, the authors could add more clarity on how methods might need to be changed to use this methodology in other settings or if there are settings where they are not appropriate to use

qRYV questions the reproducibility of the work because the authors a paid API (GPT-3) for one small part of their analysis. I don't agree that using a black-box API makes the work inappropriate for academic publication, and I don't believe the authors should be penalized for this in the absence of a higher-level ACL policy precluding comparisons with APIs.

I think the task and methodology in this work will be of interest to NLP and computational social science researchers.

---

### Decision · Program_Chairs · 2023-10-07

**Decision:**

Accept-Findings

**Comment:**

Thank you to the authors for their work and the reviewers for their reviews. This work focuses on automatically generating perspectives about abstract entities in polarized social media text, focusing on #BlackLivesMatter and #BlueLivesMatter tweets.

Pros:
- All reviewers rate the paper high for soundness
- As noted by GbZ4 "The task is challenging and interesting - identifying differing stances on the same topics - and shows up frequently in analyses of social data and social phenomena." This methodology has the potential to be broadly useful for social text analysis
- GbZ4 and Yvsx agree that the graph-based methodology with several prediction tasks is innovative

Cons:
- Several reviewers raise concerns about how the authors do not allow for neutral or ambiguous stances. As the #BlackLivesMatter movement has been highly polarized and the authors remove any users who tweet about the events <5 times, I don't see this as a fundamental flaw in this particular case study. However, the authors could add more clarity on how methods might need to be changed to use this methodology in other settings or if there are settings where they are not appropriate to use

qRYV questions the reproducibility of the work because the authors a paid API (GPT-3) for one small part of their analysis. I don't agree that using a black-box API makes the work inappropriate for academic publication, and I don't believe the authors should be penalized for this in the absence of a higher-level ACL policy precluding comparisons with APIs.

I think the task and methodology in this work will be of interest to NLP and computational social science researchers.